# Sharing water and benefits in transboundary river basins

Diane Arjoon[1], Amaury Tilmant[2], and Markus Herrmann[3]

[1,2]Department of Civil Engineering and Water Engineering, Université Laval, Québec, Canada
[3]Department of Economics, Université Laval, Québec, Canada
*Correspondence to:* Diane Arjoon (diane.arjoon@ulaval.ca)

**Abstract.** The equitable sharing of benefits in transboundary river basins is necessary to solve disputes among riparian countries and to reach a consensus on basin-wide development and management activities. Benefit sharing arrangements must be collaboratively developed to be perceived not only as efficient, but also as equitable in order to be considered acceptable to all riparian countries. The current literature mainly describes what is meant by the term benefit sharing in the context of transboundary river basins and discusses this from a conceptual point of view, but falls short of providing practical, institutional arrangements that ensure maximum economic welfare as well as collaboratively developed methods for encouraging the equitable sharing of benefits. In this study, we define an institutional arrangement that distributes welfare in a river basin by maximizing the economic benefits of water use and then sharing these benefits in an equitable manner using a method developed through stakeholder involvement. We describe a methodology in which (i) a hydrological model is used to allocate scarce water resources, in an economically efficient manner, to water users in a transboundary basin, (ii) water users are obliged to pay for water, and (iii) the total of these water charges are equitably redistributed as monetary compensation to users in an amount determined through the application of a sharing method developed by stakeholder input, thus based on a stakeholder vision of fairness, using an axiomatic approach. With the proposed benefit-sharing mechanism, the efficiency-equity trade-off still exists but the extent of the imbalance is reduced because benefits are maximized and redistributed according to a key that has been collectively agreed upon by the participants. The whole system is overseen by a river basin authority. The methodology is applied to the Eastern Nile River basin as a case study. The described technique not only ensures economic efficiency, but may also lead to more equitable solutions in the sharing of benefits in transboundary river basins because the definition of the sharing rule is not in question, as would be the case if existing methods, such as game theory, were applied, with their inherent definitions of fairness.

## 1 Introduction

With growing water scarcity, as a result of expanding population demand, environmental concerns and climate change effects, there is increased international recognition of the importance of cooperation for the effective governance of water resources. This is particularly evident in the case of transboundary river basins in which unidirectional, negative externalities, caused by the upstream regulation of the natural flow, often place some parties at a disadvantage and results in asymmetric relationships that add to the challenge of coordinating resource use (van der Zaag, 2007). There is a consensus among water professionals that the cooperative management of shared river basins should provide opportunities to increase the scope and scale of benefits

(Phillips et al., 2006; Grey and Sadoff, 2007; Leb, 2015), stepping beyond the volumetric allocation of water that reduces negotiations between riparians to a zero-sum game. In their seminal paper, Sadoff and Grey (2002) discussed the types of benefits that river basins can provide, assuming cooperation: *benefits to the river* can result from sustainable cooperative management of the ecosystem; efficient, cooperative management and development of river flow can yield *benefits from the river* in the form of increased water quality, quantity and productivity; policy shifts away from riparian disputes/conflicts toward cooperative development can reduce costs of non-cooperation arising *because of the river*; and cooperation between riparian states can lead to economic, political and institutional integration, resulting in *benefits beyond the river*.

A large proportion of past research has focused mainly on the economic benefits of cooperation (benefits from the river). Focussing on benefits in strictly economic terms does not lessen the importance of benefits from other spheres (Qaddumi, 2008). An economic perspective, however, may be an effective method for encouraging cooperation because it may help riparian countries to realize win-win situations (Dombrowsky, 2009).

The traditional approach to estimating the economic benefits of cooperation relies on hydro-economic modelling (Arjoon et al., 2014; Jeuland et al., 2014; Tilmant and Kinzelbach, 2012; Teasley and McKinney, 2011; Whittington et al., 2005). These studies present various implementation strategies representing various levels of cooperation, but all show that there are significant economic benefits to be had through basin-wide cooperation. However, economic efficiency is not necessarily compatible with equitability due to the different production abilities of water users (Wang et al., 2003). Analytical methods, including game theory solutions such as the Shapley value (Jafarzadegan et al., 2013; Abed-Elmdoust and Kerachian, 2012) and bankruptcy theory (Sechi and Zucca, 2015; Mianabadi et al., 2015; Madani et al., 2014; Mianabadi et al., 2014; Ansink and Weikard, 2012), have been examined for use in water allocation as equitable alternatives to the efficient economic allocation produced by hydro-economic models. Analytical methods were also used by van der Zaag et al. (2002) who looked at possible equitable criteria for sharing water and developed allocation algorithms to operationalize these, applying them to the Orange, Nile and Incomati rivers. It has been argued that the notion of equity, or fairness, involves a cultural component that should be incorporated into any type of water policy and, therefore, stakeholder involvement in decision-making is a significant determinant in the judgement of fairness (Syme et al., 1999; Asmamaw, 2015). The explicit provision of benefit sharing arrangements that are collaboratively developed and, thus, perceived as fair, are therefore necessary to help solve disputes and to reach a consensus in transboundary river basin development and management activities (MRC Initiative on Sustainable Hydropower, 2011).

Increasingly, efforts are focussing on the sharing of benefits generated through cooperation in order to solve the problem of equitability. The rapidly growing body of literature on benefit sharing mainly describes what is meant by this in the context of transboundary river basins and discusses benefit sharing from a conceptual point of view (Suhardiman et al., 2014; Skinner et al., 2009; Qaddumi, 2008). This literature introduces and defines different approaches but falls short of providing practical institutional arrangements for the sharing of benefits. Recently, Ding et al. (2016) introduced a methodology to address the problem of water allocation in the Nile River through a revenue re-distribution mechanism that leads to a fairly allocated revenue for each water user based on the proportion of its contribution to the basin.

35   Analytical methods, such as game theory and related bankruptcy methods, may also be useful for determining ways to fairly allocate generated benefits. Game theory, which is the mathematical study of competition and cooperation, can provide a somewhat realistic simulation of the interest-based behaviour of stakeholders (Madani, 2010). The framework that relates the preferences of players to the observable features of a game is the hypothesis that players care about nothing except their own payoffs (Hausman, 1999). Fair outcomes are captured in solution concepts such as the *core*, which selects the payoff allocations

that give each group of individuals no less than their collective worth and the *Shapley value* in which payoffs are related to the marginal contributions of individuals to a coalition (de Clippel and Rozen, 2013). The aim of bankruptcy methods is to distribute an estate or asset among a group of creditors, all having a claim to the asset, where the sum of the creditors' claims is larger that the amount available to distribute (Herrero and Villar, 2001). An overview of bankruptcy rules has been presented by Thomson (2003, 2013). Each bankruptcy rule defines fairness based on the properties underlying the rule. The three most

well-known bankruptcy rules (the proportional rule, the constrained equal awards rule and the constrained equal losses rule) all define equity through the *equal treatment of equals* requirement in which agents with identical claims should be treated the same[1]. In other words, agents with the same claim should receive the same compensation. The analysis and formulation of properties and principles of distribution rules, such as those in cooperative game theory and bankruptcy theory, is the object of the axiomatic method (Thomson, 2001).

The axiomatic method allows desirable properties to be translated into a sharing rule. If a particular rule has been adopted to solve a problem involving a group of agents, it is assumed that all agents have agreed on the properties that such a rule fulfills. The concept of fairness, then, can be embedded into a rule. The axiomatic approach is easily incorporated into negotiations because the axioms can be interpreted quite naturally as describing characteristics of a negotiation procedure (Ansink and Houba, 2014).

As discussed previously, the economically efficient allocation of water is not necessarily equitable. Axiomatic approaches, on the other hand, allow the characterization of an equitable distribution of welfare, but do not necessarily maximize the aggregated economic welfare over the basin. Institutional arrangements that ensure maximum economic welfare, as well as the equitable sharing of these benefits over the basin, are required.

In this study we define an institutional arrangement that distributes welfare in a river basin by maximizing the economic

benefits of water use and then sharing these benefits in an equitable manner. The methodology relies on a pseudo-market arrangement in the form of a highly regulated market in which the behaviour of water users is restrained to control externalities associated with water transfers and to ensure basin-wide coordination and enhanced efficiency. The term pseudo-market indicates that bulk water users are not free to choose how much water will be moved in the system. Freedom of contract and private property rights, which are necessary conditions for the existence of a market, are restrained, giving rise to a pseudo-market[2].

These restrictions are due to the flow characteristics of water and to the need to account for externalities and third-party effects, which can seldom be achieved within a traditional market.

---

[1]Equal treatment of equals is one of the properties upon which these bankruptcy rules are defined. For a complete discuss of all properties, refer to Thomson (2003, 2013).

[2]One could also argue that a true market is created by assuming that every agent agrees with, and respects, having to pay for water.

The institutional arrangement described in this paper should encourage full cooperation between water users because it is intended as a replacement for traditional types of agreements on international river basins, which can lead to distrust and tension between riparian countries. What we present is an entirely different perspective that may help to avoid the pitfalls and limitations of current agreements.

In the following section, we describe this arrangement which uses a hydro-economic model to determine the economically efficient allocation of water and a collaboratively developed sharing method for the equitable allocation of monetary benefits. Section 3 presents the application of this framework to the Eastern Nile River basin. Section 4 presents and discusses the results and Sect. 5 concludes the paper.

## 2 Methodology

In the proposed pseudo-market approach, a river basin authority (RBA) plays the role of water system operator, identifying economically efficient allocation policies which are then imposed on the agents (water users). The agents are charged for water use and these payments are redistributed to ensure equitability among the users. In this particular system, the mandate of the RBA consists of (1) collecting information on water use and productivity, (2) efficiently allocating water between the different agents in the system based on the information collected in the first step, (3) preserving the hydrologic integrity of the river basin, and (4) coordinating the collection and redistribution of the benefits associated with the optimal allocation policies.

### 2.1 Information Collection

In this first step, the RBA collects information that is required to assess the demand curves, or at least the productivity (unit net benefit), of all users in the system, once at the beginning of each year. The information must be validated to ensure that it is complete and reasonable since the economically efficient allocation of water in the next step depends on it. The collection of information can be the basis of a bidding process in which agents offer to buy water at a given price. In the case of irrigation agents, information such as crop area, crop type, yield, crop price and crop water requirement over a period can be used to determine the bid for each agent and, based on the bid information, the demand curve can be inferred using the residual imputation method (Pulido-Velazquez et al., 2008; Riegels et al., 2013). This method assumes that all input costs, except for the cost of water, are known. The water value is then imputed as the residual of the observed gross benefits after all non-water costs are subtracted (Young, 2005).

In order to control the declarations of agents in the agricultural sector, the RBA can use techniques such as remote sensing to validate land classification and cropping areas (Gallego et al., 2014; El-Kawy et al., 2011; Rozenstein and Karnieli, 2011). As an example, the European Union uses an Integrated Administration and Control System (IACS), which includes a land-parcel identification system (LPIS), to control declarations from farmers for financial aid grants (Oesterle and Hahn, 2004). The LPIS uses orthophotos to monitor the evolution of the land cover and the management of crops, and enables more accurate declarations by farmers.

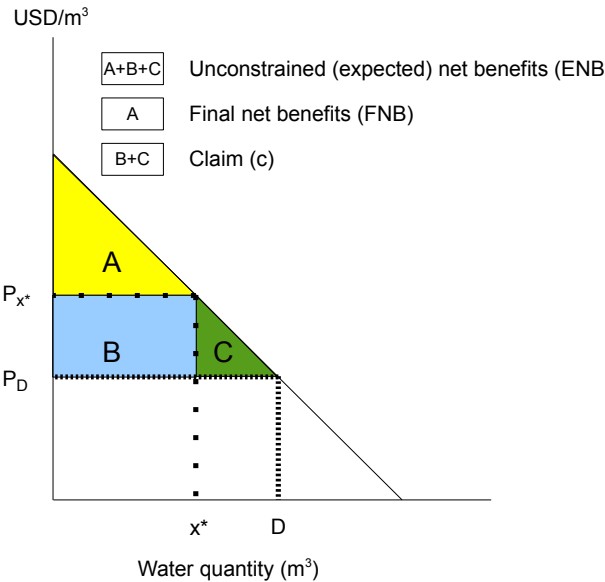

**Figure 1.** Demand curve. D=quantity of water demanded for a time period, $x^*$=quantity of water allocated for a time period, P=price of water.

In the case of hydropower, information regarding energy production and scheduling is important. For example, power plants might be off line for maintenance or might be obliged to generate a minimum amount of energy to meet its contractual commitments. As well, water use requirements such as environmental flow and minimum domestic use supply will also be required.

The unconstrained or expected net benefits (ENB) for a water user is the consumer surplus (Fig. 1), which is the area under the demand curve above the price $P_D$. The surplus is the private user cost of water and corresponds to the willingness to pay for the last unit of water demanded in a situation where allocation is unconstrained. This area is made up of three regions (A, B and C) which will be discussed later.

## 2.2 Water Allocation

Once water user information has been collected, allocation decisions are identified by matching demand with supply in a cost-efficient way, i.e. by giving priority of access to users with the highest productivity. In order to do this, an aggregation of the demand curve is carried out, which means that a distinction must be made between rival and non-rival water uses. When water users are not in competition for the same unit of water, non-rivalness is observed. For example, water flowing through a dam may be considered a non-rival water use since a unit of water released through one dam can be used downstream by another dam. In rival water use, units are consumed and are no longer available to other water users (for example, water lost to irrigation or water held in a reservoir during a period when it is required downstream for irrigation). In this case, the demand curves are summed horizontally (see Fig. 2). Rival water uses need to be coordinated to prevent conflicts. The decision to divert

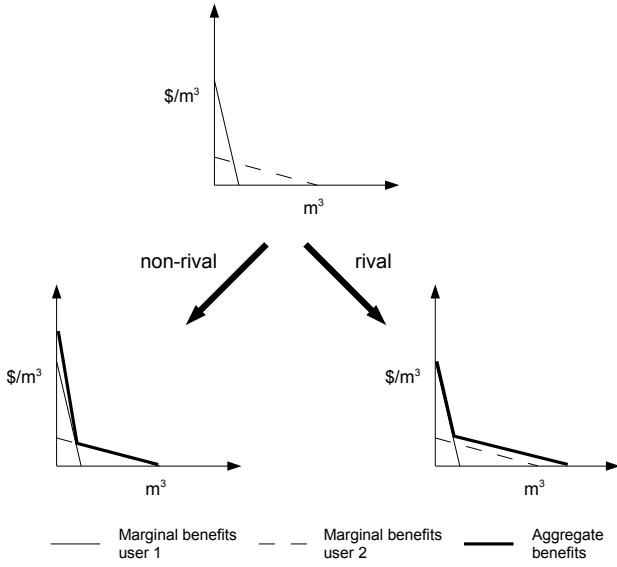

**Figure 2.** Aggregation of demand curves for rival and non-rival water uses for a given time period.

one additional unit of water to any rival use depends on the at-source value[3] of water for that use. If this value is larger than the at-source value of all downstream marginal users, then it will be diverted to the rival use. See Tilmant and Kinzelbach (2012) for a detail description of rival and non-rival water uses. The value of the last unit of water at any site, then, is the sum of the marginal values of the non-rival users since the demand curves can be summed up vertically (see Fig. 2). This aggregation of the demand curve is done automatically in hydro-economic models. Hydro-economic models, then, can be used to determine the allocation of water between users at the same site and over a basin (comprising a number of sites) and to determine the marginal value of water and economic benefits at each site. A description of the mathematical formulation involved is given in the Appendix.

## 2.3 Collection of Bulk Water Charges

Based on the water allocation decisions and the corresponding water fluxes, pseudo-market transactions occur between the RBA and the water users. Users must pay the RBA for the water allocated to them. The cost of water is the marginal water value or shadow price ($\lambda$) calculated by the hydro-economic model at the site of water abstraction or use. Economic theory indicates that for efficient water allocation to occur, the price that users pay for the resource must be equal to the marginal value of still available opportunities of water use, which reflects the social cost of using water at a particular site. If the user pays less

---

[3]The at-source value of water is observed at the location where bulk water is diverted. The at-site value corresponds to the value of water delivered to the users (for example, a farm at the end of a conveyance and distribution system). At-site water values are generally larger than at-source values because they include losses in the system and conveyance costs. In the study of intersectoral allocation choices, at-source water values should be used (Young, 2005).

than this, the resource is over consumed or over utilized, as no efficient rationing occurs. Conversely, a user price higher than the marginal value would result in underconsumption/underutilization.

The RBA charges for the water entering the system in order to cover the costs associated with its mandates (conservation, coordination, compensation). In the case of consumptive users, water is purchased from the RBA at the marginal water value (the value of a marginal unit of water) at the site of abstraction. Non-consumptive users buy inflow from the RBA at a price equal to the difference between the marginal value of water at the user site and the marginal value of water at the downstream site (Fig. 3). This bulk water charge system is based on a dynamic water accounting framework presented by Tilmant et al. (2014).

Payment for bulk water use has been addressed, recently, by the United Nations in their 2014 World Water Development Report (United Nations World Water Assessment Programme, 2014) in which they state that economic instruments such as markets for buying and selling a resource (such as water) or the imposition of water use tariffs could create incentives for more efficient use. And, in fact, payment for bulk water supply has been established in recent water laws in Zimbabwe, Tanzania and Mozambique (The World Bank, 2008).

Once transactions are collected by the RBA, water costs ($CW$) for each water user can be calculated along with the final net benefits ($FNB$) which is equivalent to the consumer surplus shown, on Fig. 1, as the area above the line $P_{x^*}$ (area A). Line $P_{x^*}$ is the social cost of water where $x^*$ is the economically efficient water allocation.

The difference between the amount of benefits expected by each agent ($ENB$) and the final net benefits received ($FNB$) is the amount an agent will claim for compensation in the next step ($c$) and is equal to the value of the externalities (B+C on Fig. 1). These claims are composed of the difference in water costs between the unconstrained water demand ($D$) and the actual water allocation ($x^*$), which is area B on the figure, and the cost of cooperation ($CC$) which is the loss in benefits due to the allocation of less resource than what was demanded (area C in the figure).

## 2.4 Transfer Payments

At this point in the methodology, the RBA has collected an amount of money, referred to as the *estate* ($E$), that can be shared among the water use agents. Using an axiomatic approach, a method of sharing this estate should be determined. The aim of the axiomatic approach is to find and capture the notion of fairness that water users could agree upon. The approach then sets out axioms (properties) that fairness should or should not satisfy. Finally, these properties are translated into a sharing rule that quantifies the particular definition of fairness. How the benefits are shared depends entirely on this definition as agreed to by water users. For example, a simple proportional sharing method may satisfy the properties of equity defined by the users, or an egalitarian method, or some other form of sharing may be required. Since each river basin will have a different definition of fairness (depending on conditions in the basin and the outcome of negotiations with the water users), each river basin will likely have its own unique sharing rule.

A flowchart of the complete methodology, including information obtained at each step, is shown in Fig. 4.

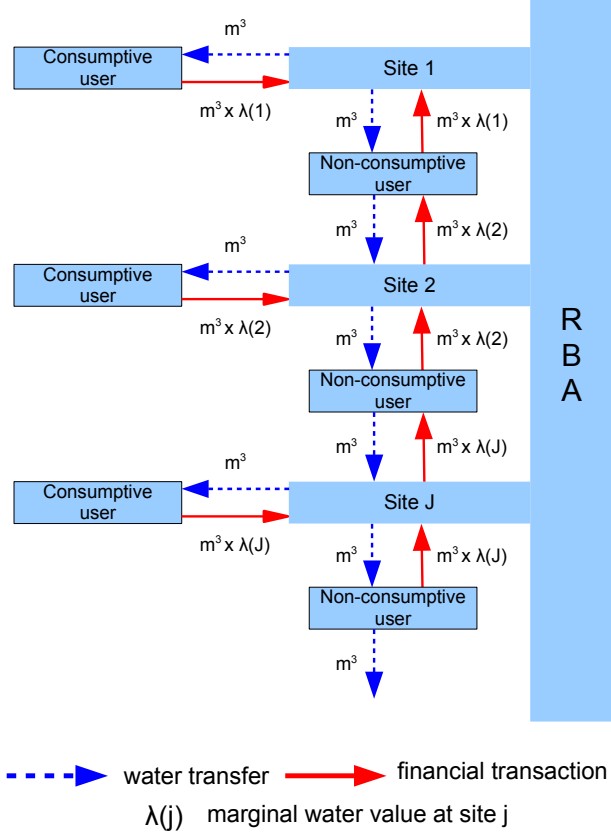

**Figure 3.** Collection of bulk water charges for a given time period.

 ## 3   Case Study

### 3.1   Eastern Nile River Basin

The Eastern Nile River basin is used to illustrate the methodology described in the previous section. Covering an area of approximately 330,000 km$^2$ and with a length of 1529 km, the Blue Nile originates in the highlands of Ethiopia and flows into Sudan where it joins the White Nile at Khartoum to form the Main Nile. The Main Nile then flows out of Sudan, into Egypt and discharges into the Mediterranean Sea. The Eastern Nile River basin is composed of the Blue Nile, the Tekeze-Atbara, the Baro-Aboko-Sobat, the White Nile downstream from Malakal and the Main Nile sub-basins (Fig. 5).

The dominant uses of water in the Eastern Nile River basin are irrigated agriculture and hydropower generation, mostly in Sudan and Egypt. This is, however, likely to change in the near future with the completion of the Grand Ethiopian Renaissance Dam on the border of Ethiopia and Sudan.

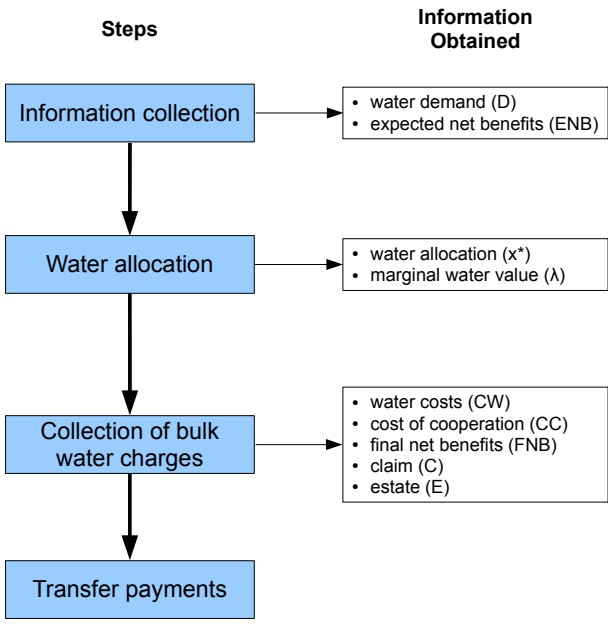

**Figure 4.** Flowchart of methodology including information obtained at each step.

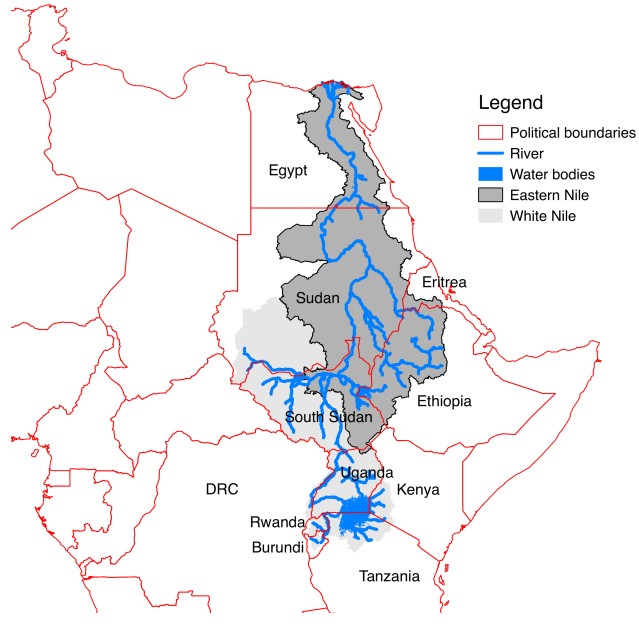

**Figure 5.** Eastern Nile River Basin

There is a long history of unsuccessful negotiations over water allocation and development of Nile water resources. Attempts at cooperation and benefit sharing within the Eastern Nile basin go back to the early part of the 20th century. The 1929 Nile Waters Agreement between Sudan and Egypt prioritized Egyptian water needs and reportedly gave Egypt the right to veto future hydroelectric projects along the Nile (Brunnée and Toope, 2003). Sudan and Egypt subsequently replaced the 1929 treaty, in 1959, with the Agreement for the Full Utilization of the Nile Waters, which essentially allocated the entire flow of the Nile at the Aswan Dam to Sudan and Egypt. Unsurprisingly, this has caused regional tension with the other riparians, who invoke the Nyerere Doctrine[4], and general principles of international water law, to contest the 1959 Agreement and claim a share of the Nile waters.

In 1999 the Nile Basin Initiative (NBI) was undertaken with the goal being to adopt a comprehensive, permanent, legal and institutional agreement over the Nile River basin. So far there has been little success in negotiations leading to an agreement. However, a Cooperative Framework Agreement (CFA) was signed by a number of the Nile basin countries with the notable exceptions of Egypt, Sudan and South Sudan.

Regional tensions have further complicated Nile cooperation efforts. For example, Ethiopia and Egypt have a long history of distrust and Egypt and Sudan, as well as Eritrea and Ethiopia, have long unresolved border disputes. Additionally, many Nile riparians have been broken by internal conflicts and instabilities that result in challenges to international relations.

In recent years, the construction of the Grand Ethiopian Renaissance Dam has been a source of concern and conflict among the three riparian countries. It should be noted, however, that in early 2015, Egypt, Sudan and Ethiopia signed an agreement on the declaration of principles with respect to the project.

It is pretty much agreed, at this point, that benefit sharing may offer a solution to the stalemate surrounding water use and allocation in the Eastern Nile River basin. While the concept of benefit sharing can be appreciated by most riparian countries, questions regarding methods of sharing benefits have emerged. The three Eastern Nile River basin countries need to, first and foremost, identify the bundle of benefits that can be generated, then agree on a mechanism for sharing these (Tafesse, 2009)).

## 3.2 Information Collection

Given the lack of accurate data with respect to irrigated agriculture in the Nile River basin, a net return of 0.05 USD/$m^3$ is chosen (as in Whittington et al. (2005)). For hydropower it is assumed that each MWh generated has an economic value averaging 80 USD/MWh for firm power and 50 USD/MWh for secondary power. These values are consistent with feasibility studies of hydroelectric dams in Ethiopia. Using these values the unconstrained expected net benefits (ENB) are determined for each water use agent as:

$$ENB_j = D_j * P_j \tag{1}$$

---

[4]The Nyerere Doctrine of state succession, founded by the first President of Tanzania, states that a new nation should not be bound to international agreements dating back to colonial times and that these agreements should be re-negotiated when a state becomes independent.

where $D_j$ is the unconstrained quantity of water demanded by agent $j$ and $P_j$ is its productivity. Note that the assumption is made that users do not currently pay for water.

The water demand for the irrigation agents is equal to the crop water demand. For the hydropower agents the water demand is equal to the amount that they are allocated in the next step. Since the allocation is economically efficient, the hydropower agents are assumed to be satisfied with the amount of water flowing through the turbines.

### 3.3 Water Allocation

The stochastic multistage decision-making problem (Eqs. (9) to (12) defined in the Appendix) was solved using stochastic dual dynamic programming (SDDP). Details of this algorithm can be found in Goor et al. (2010) and in Tilmant and Kinzelbach (2012). The hydro-economic model of the Eastern Nile basin is based on the schematization shown in Fig. 6. In this study the assumption is made that the Grand Ethiopian Renaissance Dam (located at H8 in Fig. 6) is online. Allocation decisions are chosen to maximize expected net economic returns from irrigated agriculture and hydropower generation over a planning horizon of 10 years and for 30 hydrologic sequences (see Arjoon et al. (2014) for a description of the model).

Once the allocation decisions are determined, the actual gross benefits (GB) can be calculated as:

$$GB_j = x_j^* * P_j \tag{2}$$

where $x_j^*$ is the water allocation decision for agent $j$. The difference between the expected net benefits (ENB) and GB is the cost of cooperation (CC) to the agent due to the efficient allocation of water. In other words, it is the difference between the amount of benefits the agent is expecting to get if their unconstrained water demand is met and the actual benefits the agent receives given the allocation decision, excluding water costs.

### 3.4 Collection of Bulk Water Charges

The total of the transactions collected by the RBA ($E$), minus yearly operating expenses of 3 million USD, will be used to compensate the agents for a percentage of the benefits lost either through efficient allocation (cost of cooperation) or water costs. Operating expenses of 3 million USD/yr is in line with those published by power pools (Southern African Power Pool, 2009) and river commissions (Mekong River Commission, 2013).

Final net benefits for each agent can be calculated as:

$$FNB_j = GB_j - CW_j \tag{3}$$

where $CW_j$ is the cost of water for agent $j$.

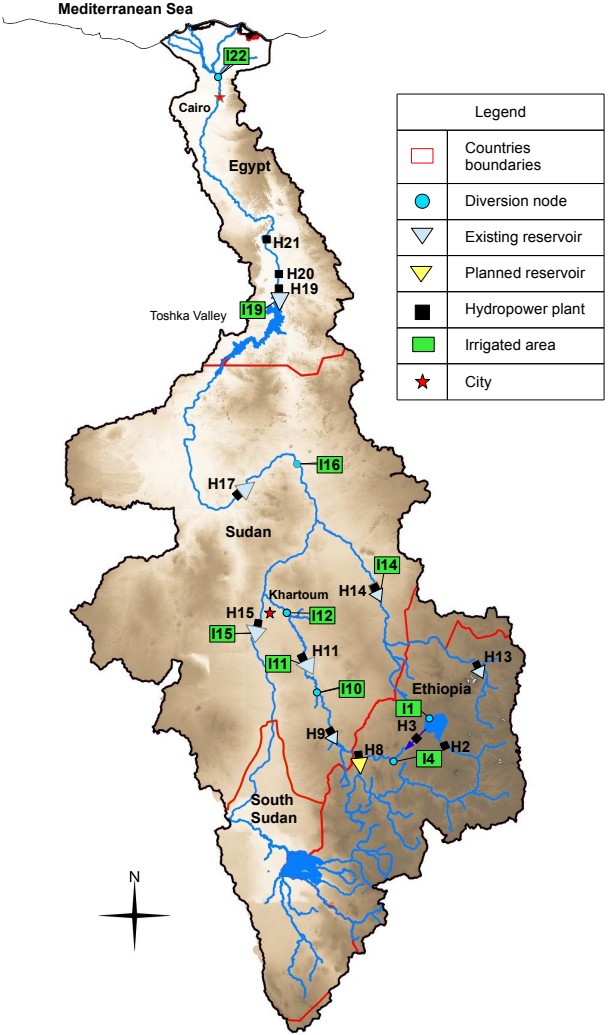

**Figure 6.** Model schematic of the Eastern Nile River basin. Irrigation agents (I) and hydropower agents (H) for this case study are shown. Note that the numbering is not consecutive because there are nodes that represent agents that are not part of the case study.

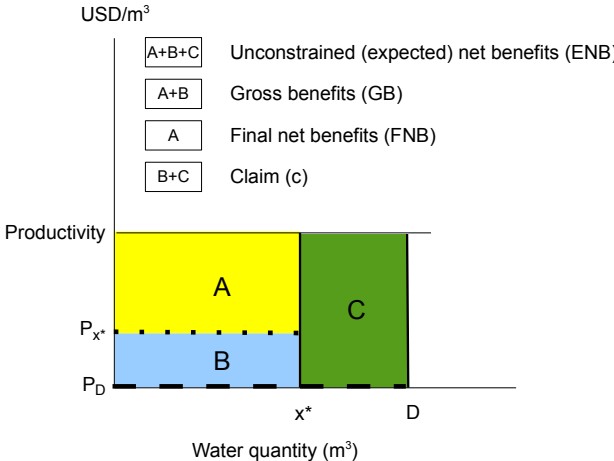

**Figure 7.** Demand curve for the case study. D=quantity of water demanded for a time period, $x^*$=quantity of water allocated for a time period, P=price of water.

## 3.5  Transfer Payments

Once the final net benefits have been determined, transfer payments can be calculated for each agent. To do this, the total cost for each agent needs to be calculated, which will give the upper limit to the claim (c) of an agent to the estate.

Fig. 7 shows the annual demand curve for an irrigation agent in this case study. In this study, we implicitly assume that the input demand is horizontal (perfectly elastic) with the price (P) = marginal productivity. The area to the left of line D (comprising areas A, B and C) is the expected net benefits (ENB) (we see that the agent does not pay for water) resulting from unconstrained water use. When water is constrained, area A is the final net benefits (FNB). The claims (c) are divided into two parts: area B is the cost of water (CW) to the agent and area C is the cost of cooperation (CC) due to the efficient allocation of water. Area B also represents the amount of money that the RBA collects from this agent. As previously mentioned, for hydropower agents the water demand and the water allocation are equal, therefore there is no cost of cooperation. The claim (c), then, for a hydropower agent, is the cost of water (CW). Over the whole basin the amount that the RBA collects (and is available for transfer payments) is enough to reimburse the agents for the actual cost of water, however, as mentioned, 3 million USD is held back for annual operating expenses. Therefore the shortfall between the amount the RBA has to share and the claims of the agents is the total cost of cooperation for irrigation agents ($\sum CC_j$) plus operating expenses.

The situation in which the amount available to share between agents is less than the total claims of the agents is, by definition, a bankruptcy problem.

In this case study, the collected benefits are shared among the water use agents following a rule that was developed based on a number of well defined properties in the bankruptcy literature (*feasibility, non-negativity, claims-boundedness*) as well as some that are specific to the problem (*solidarity, security of minimum benefits*).

It should be noted that, for this study, the properties of this rule were not developed with stakeholder input, as this was beyond the scope of this research project. Although stakeholder involvement is imperative in this institutional arrangement, in this case study, we are giving an objective viewpoint and this analysis serves as a benchmark or reference point.

Benefits are shared in such a way as to ensure that each agent has the same proportion of final costs $(ENB_j - (FNB_j + tp_j))$ to benefits demanded $(ENB_j)$ (where $tp_j$ is the monetary transfer payment made to the agent) and that these are minimized. By extension, this rule also ensures that each agent receives an equal proportion of final benefits $(FNB_j + tp_j)$ to benefits demanded $(ENB_j)$ and that these are maximized. This rule also applies a *solidarity* property in which all agents take equal responsibility for the shortfall in benefits at certain nodes due to the efficient economic allocation of water over the basin, and a property of *security of minimum benefits* in which the benefits obtained from the use of water $(FNB_j)$ are uncontested.

The compensation rule is defined as follows :

$$tp_j = ENB_j - (FNB_j + \gamma ENB_j) \tag{4}$$

where $\gamma$ is chosen such that :

$$\sum tp_j \le E \tag{5}$$

Equation (5) ensures the property of *feasibility* which is the requirement that the sum of the transfer payments not exceed the amount available to share.

The following constraints also apply :

$$tp_j \ge 0 \tag{6}$$

$$tp_j \le c_j \tag{7}$$

Equation (6) ensures *non-negativity*, which requires that each agent receive a non-negative amount, and Eq. (7) ensures *claims boundedness* which requires that each agent receive, at most, the amount of its claim.

Rewriting Eq. (4) to read

$$\gamma = (ENB_j - (FNB_j + tp_j))/ENB_j \tag{8}$$

shows that the property of *solidarity* is supported by ensuring that the final cost $(ENB_j - (FNB_j + tp_j))$ to expected benefit $(ENB_j)$ ratio for all agents is the same.

In this final step, the transfer payments are calculated and the total final benefits $(FNB + tp)$ for each agent are determined.

## 4 Results

25   The analysis of results was carried out on year 4 of the 10 year planning horizon. This ensures a steady-state condition that is not influenced by initial hydrological and storage conditions or by any end-effect distortion due to reservoir depletion that occurs as the end of the planning period approaches (Arjoon et al., 2014). As previously explained, the amount of water allocated to hydropower agents is equal to the amount demanded. This means that all hydropower agents receive 100% of the water demanded. The efficient allocation of water results in most irrigation agents also receiving their unconstrained demand. The exceptions are agents I1, I4 and I14 who receive, on average, 1%, 0% and 94% of their unconstrained demand, respectively (See Fig. 8) . This result is not unexpected because, from an economic standpoint, irrigation in the Eastern Nile River basin should take place downstream after water has been used for hydropower generation upstream (Whittington et al., 2005). These

5   three irrigation agents have cooperation costs as well as, possibly, water costs. Looking at the cumulative distribution of the proportion of the allocated amount of water to the amount received for these agents (Fig. 9) we see that 95% of the time, agent I1 does not receive any water. Agent I14, on the other hand, receives its full demand about 75% of the time. Agent I4 (not shown in Fig. 9) always receives 0%. The rationing of water for upstream irrigation users is a result of the horizontal demand curve used for irrigation. If more detailed economic/agricultural data were available, a non-horizontal demand curve could be

10   produced. This may result in irrigation schemes with high value crops having priority to water and those areas with low value crops not being irrigated. This means that the irrigation water users that are rationed may change and they may be more spread out over the basin.

  Overall, the agents with the smallest claims are all hydropower agents in Sudan (H9, H11, H14, H15) with marginal values that are almost equal to marginal values at the downstream sites (see Fig. 10). This means that they sell water downstream at

about the same price that they paid for it, resulting in lower water costs. Figure 11 gives a basin-wide view of the percentage of the unconstrained benefits claimed by each agent, by agent type, on average. The irrigation agents upstream claim a larger percentage of their expected benefits because, first, they pay more for water and, second, they also have cooperation costs. With respect to hydropower agents, H8 and H19 (Grand Renaissance and High Aswan, respectively) claim the largest percentage of their expected benefits. In both cases, the cost of water at these sites are much greater than the cost of water at the respective

downstream sites.

  From the collection of bulk water charges for the period analyzed (year 4), the RBA ends up with 3894 million USD to allocate between the agents (after removing 3 million USD for operating costs). The total claims for all agents, for the year, is 4266 million USD which means that there is a shortfall of 372 million USD between the amount available to share and the claims, or about 9% of the total claims.

Using the bankruptcy rule developed for this example, the average amount of transfer payment is calculated for each agent. The ratio of final net benefits (FNB) to expected net benefits (ENB), referred to as the *initial ratio*, and final net benefits plus transfer payments (FNB+tp) to expected net benefits (ENB), referred to as the *final ratio*, are determined and analyzed. These results were analyzed over the 30 different hydrologic sequences to assess how this rule performs under varying hydrologic conditions.

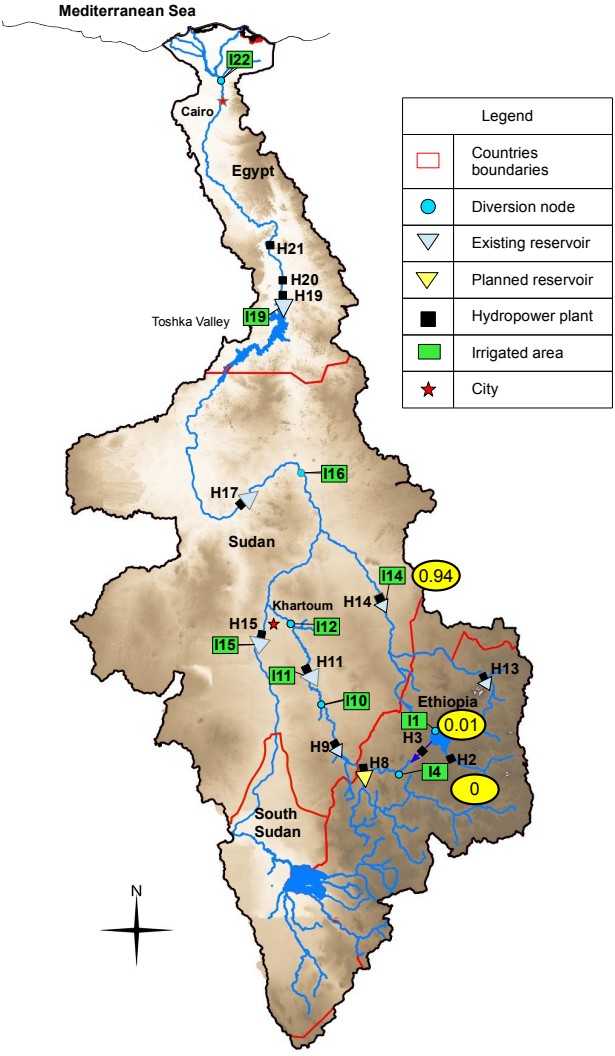

**Figure 8.** Average proportion of water allocation to unconstrained demand for all agents. Only the values for those agents in which the proportion is less than 1 are shown.

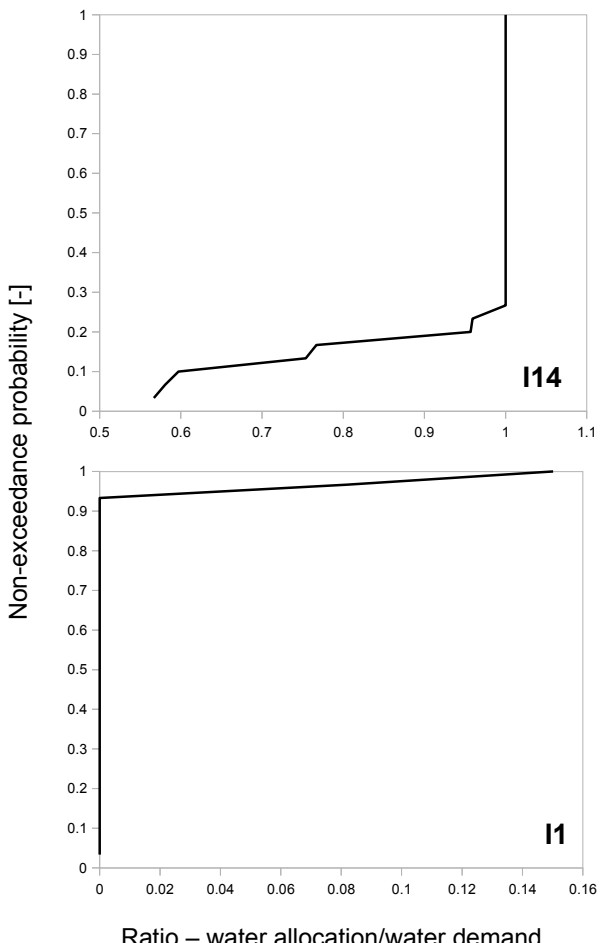

**Figure 9.** Cumulative distribution function for the proportion of water allocation to unconstrained demand for agents I1 and I14.

Figure 12 shows the mean values for initial ratios (shown as large filled squares) and final ratios (shown as large filled diamonds) for irrigation agents as well as the values for each of the hydrologic sequences. Agents I1 and I4 receive little or no irrigation water, on average, as discussed previously. Agent I14 initially receives about 23% of its expected net benefits, on average. This agent is located at the Kashm El Girba dam, on the Tekeze-Atbara River. The flow of this river is highly seasonal with annual flows entering Sudan from Ethiopia restricted to the flood period of July to October. The design storage capacity of the reservoir at this site is about 10% of the inflow, however, high sedimentation in the reservoir dropped the storage capacity by 50% as of 1977. This loss of storage capacity has resulted in severe water shortages during drought years and an associated decline in the crop area cultivated. As a result, the restriction of water for this irrigation agent is more probably due to the hydrology as opposed to being economic in nature. Due to flow variation, the marginal water values are highly variable at this site, resulting in a wide spread of initial ratios over the hydrologic sequences (as indicated by a large vertical spread of data

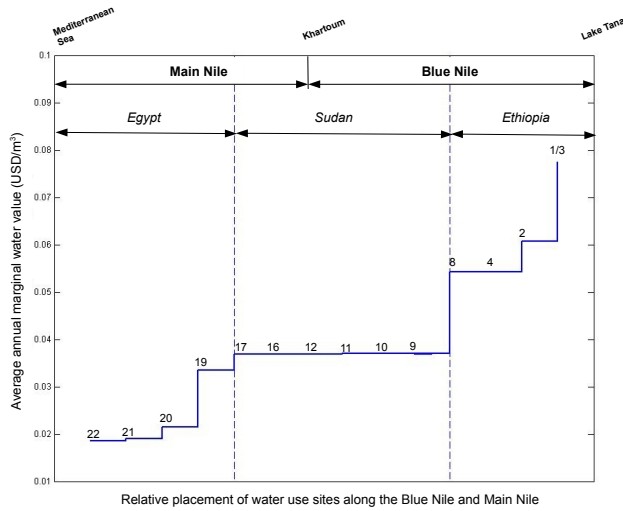

**Figure 10.** Marginal Water Value - Blue Nile and Main Nile

10    points on the graph for this agent). All other agents always receive their full unconstrained demand. Variability in the initial ratios of these agents are due to variability in the marginal water values over the hydrologic sequences.

Results for hydropower agents are shown in Fig. 13. Here we see more variation in the initial ratio than for the irrigation agents. The upstream hydropower agents (H2, H3), and those on the Tekeze-Atbara River (H13, H14) have large variations in initial ratios as a result of large inter- as well as intra-year variations in flow (and subsequently in marginal water values) which

15    occurs because these sites are all upstream of flow regulating infrastructure. The agents with the smallest claims are the four smallest hydropower agents in Sudan (H9, H11, H14, H15). These agents have the largest initial ratios and, therefore, often do not receive monetary transfers. This also results in the final ratios for hydropower agents not being equal because the property of non-negativity, which is used to define the sharing rule, allows an agent to keep its initial benefits from water use even if this results in its final ratio being larger than those of the other agents.

Overall, the average final ratios for all agents (irrigation and hydropower) are equal, with the exception of agents H9, H11, H14 and H15 as mentioned above. There is also very little variation in final ratio values with respect to hydrologic sequence. The final ratio for irrigation agents varies from 93.5% to 95% of their uncontested benefits. For hydropower agents the statistical distribution of final benefit ratios is shown in Fig. 14. We see that these final ratios also vary between 93.5% and 95% with

the exception, again, of agents H9, H11, H14 and H15 which have high initial ratios that vary with inter- and intra-annual variations in the marginal value of water. These results indicate that the sharing rule used is predictable in that agents can expect similar final benefits regardless of the hydrologic conditions.

Results that warrant a closer look are those for the upstream irrigation agents I1 and I4. We can conclude that, in this case study, given the economic information used in the model, it is economically inefficient to irrigate upstream in the basin

regardless of the hydrologic sequence (meaning that even in situations of high flow years, there is no irrigation water allocated

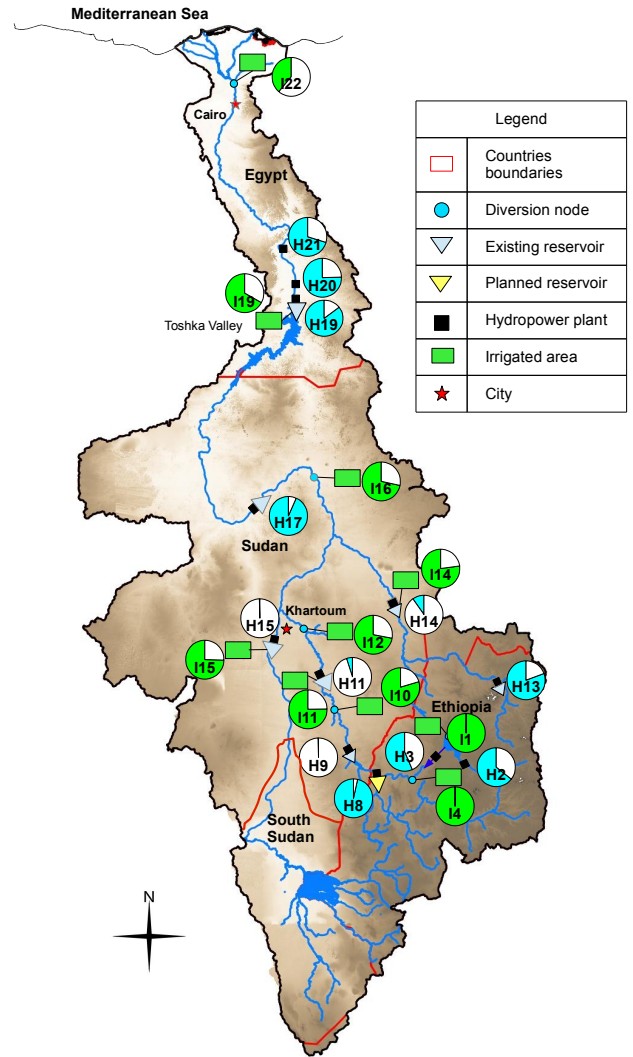

**Figure 11.** Percentage of unconstrained benefits claimed by agents.

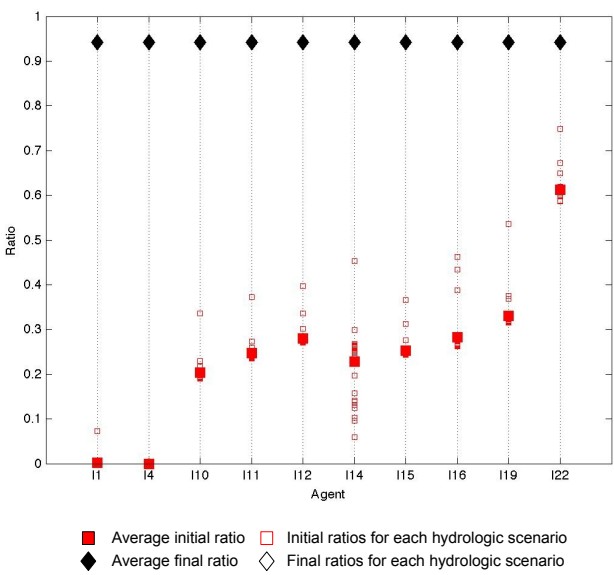

**Figure 12.** Initial and final ratios for irrigation agents.

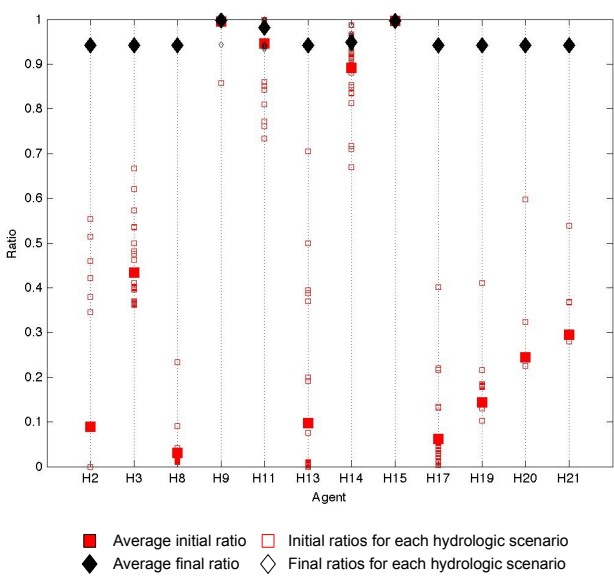

**Figure 13.** Initial and final ratios for hydropower agents.

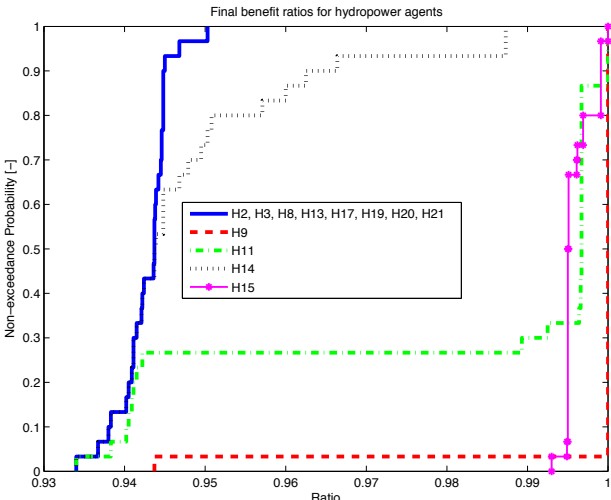

**Figure 14.** Final benefit ratio for hydropower agents.

to these agents). However, these two irrigation agents consistently demand fairly substantial transfer payments even though they do not contribute economically to the basin. This becomes an obvious problem of fairness for the other agents. If these results persist over a number of years, the RBA could use this information for better management by ensuring that agriculture is developed downstream or that upstream agricultural sites have a high productivity value.

Finally, it should be noted that we make no attempt to compare the results of the case study with current water use in the basin. While the presented case study is hypothetical and is not consistent with the actual, current situation, it represents a possible long-term future scenario in the basin and the results reflect these assumptions. In the case study, we assume complete cooperation, there is expanded irrigation in the basin and the Grand Ethiopian Renaissance Dam is online.

## 5   Conclusions

The sharing of benefits among agents in a transboundary river basin is based on three fundamental questions : (i) how can the benefits of water use be quantified and monetized, ii) what mechanism can be used to allocate benefits, and (iii) upon what criteria should the sharing of benefits be based to ensure efficiency and equitability. It should be noted that there is no unique response to these questions. In this paper, we propose one approach for distributing the benefits of cooperative management in a river basin system comprised of rival and non-rival uses. To illustrate the approach, we used the Eastern Nile River basin as 25    a case study due to the important hydropower and agricultural sectors spread over three countries.

The methodology described in this paper is based on the welfare distribution for each agent being equal to the sum of its benefits from water use plus a monetary transfer. First, efficient water allocation is implemented through the application of a hydro-economic model in order to maximize the benefits in the river basin. Second, a charge for the use of water is established. The price that agents pay for the use of water is equal to the marginal value of water at the site at which the agent receives its

allocation. The total of the water charges is equivalent to the overall value of water in the basin that is used in the sectors being studied. Finally, the total of the water charges are reallocated over the basin to ensure that all agents pay the same ratio of costs to benefits, using an axiomatic approach. The whole system is overseen by an RBA.

The two main goals of benefit sharing, efficiency and equitability, are the foundation of this methodology. The hydro-economic model results are the efficient water allocations for each agent. Efficiency is also inherent in the benefit sharing rule used to implement the monetary transfers in that all of the available money is shared among the agents. The defined properties of fairness are embedded in the sharing rule through the axioms.

This methodology can be useful to policy-makers in that the solution is more likely to be perceived as equitable, resulting in water use agents being more open to cooperation. An additional advantage of this method is the predictability of the final results. These results, over varying hydrological sequences, are shown to be relatively constant.

The importance of this methodology is that it can be adopted for application in negotiations to cooperate in transboundary river basins. The methodology is flexible in that there is no set way to allocate the water over the basin. Any hydro-economic model (or another method) can be used as long as the amount of water allocated to each agent, as well as the marginal value of water for each agent, is available. As well, the development of the sharing rule can be based on stakeholder input and will depend on specific conditions in specific river basins.

One obvious constraint of this method is its dependance on the existence of a strong basin-wide authority to impose fees and that can enable negotiations between stakeholders for the development of a sharing rule. Allowing all stakeholders a place at the table might prove challenging, especially for large systems with diversified water use activities. In the irrigation sector, for instance, farmers could be represented by a water user association. For uses of water as a public good, such as for environmental flows, the representative could be the Ministry of Environment of the country of interest. For municipal uses, the system could be designed in such a way that a minimum amount of allocated water is guaranteed (a fixed constraint in the allocation system) while quantities beyond that minimum would be part of the pool for which municipalities would have to bid. Industrial and power companies are easier to handle. All users that can be rationed (mainly private water users) are allowed a place at the table for the purpose of defining fairness with respect to transfer payments. Another possibility is that the government (or at least a high level representative of the stakeholders) has the ultimate negotiation power, akin to negotiations on trade liberalizations. Clearly, different lobbies exist that would try to influence the government, implying, ultimately, some form of compensation (the analysis of which is outside the scope of this paper).

Another constraint is the availability of reliable data. Some information such as market prices, either national or international, can be observed and transportation costs can be estimated, allowing for an approximation of the mark-up that may accrue to farmers, for example. This paper describes a system in which it is assumed that there is cooperation over the whole basin and that water users have agreed to bid for water and to supply the information that is necessary to make the methodology work. Increasingly, the information required is becoming available through the use of remote sensing and monitoring of river basins.

Incentives for water users to cheat, with respect to the data they provide, will remain even if the river basin authority is able to audit the bids. For industrial uses, including hydropower generation, cheating might be more difficult because the market prices and production functions are often well characterized. The main challenge is to be found in the agricultural sector

because (a) it is often the largest water use in a basin (and, hence, cheating might have serious basin-wide consequences), and (b) the heterogeneity in terms of cropping patterns and irrigation efficiency requires that significant data be collected and analyzed to audit the demands. We argue that the incentives to cheat might not be eliminated but they can be suppressed, or at least kept within limits, through a robust monitoring system and a strong RBA to negotiate disputes. An example of how this has worked, with good success, is the Indus River basin. Zawahri (2009), in discussing the Permanent Indus Commission, states "The commission's ability to monitor development of the shared river system has permitted it to ease member states' fear of cheating and confirm the accuracy of all exchanged data. Finally, its conflict resolution mechanisms have permitted the commission to negotiate settlements to disputes and prevent defection from cooperation."

This paper adds to the analysis of the sharing of economic benefits in transboundary river basins by describing a methodology for efficient and equitable benefit sharing based on operating the river basin as a water pseudo-market with the advantages of resource use optimization, improved resource reliability and enhanced security of resource supply. As well, we impose specific axioms, based on a stakeholder vision of fairness, on the compensation scheme and derive a unique solution for the distribution of monetary payments. This technique may lead to a sharing solution that is more acceptable to shareholders because the definition of the sharing rule is not in question, as would be the case if we applied existing bankruptcy rules or other game theory solutions with their inherent definitions of fairness.

# 6  Appendix

Hydro-economic modelling is a common tool used to analyze river basin systems and, specifically, water resources allocation problems. These models use a network representation of the system in order to physically connect various sources of supply with scarcity-sensitive water demands. Reviews of hydro-economic models can be found in Harou et al. (2009) and Brouwer and Hofkes (2008). Two classes of hydro-economic models exist: optimization-based and simulation-based. Both approaches have advantages and disadvantages but the allocation decisions and the marginal costs of the binding constraints (the limiting resources or factors that prevent further improvement of the objective function) determined by an optimization model makes this type of model attractive in the proposed methodology. In the system network, a water balance is evaluated at each node to determine the amount of water available for the demand sites connected to that node. The mass balance equation ensures that water is allocated to the connected water users to the extent permitted by water availability at the node. In the case of water scarcity, the marginal cost associated with the water balance indicates the shadow price of water or what the users would be willing to pay for an additional unit of water (Young, 2005).

In a hydro-economic water resource optimization problem, the objective function $Z$ to be maximized includes the economic net benefits across all water uses over a given planning period.

$$Z^* = \max_{x_t} \left\{ \mathop{\mathbb{E}}_{q_t} \left[ \sum_t^T \alpha_t b_t(\mathbf{w}_t, \mathbf{x}_t) + \alpha_{T+1} \nu(\mathbf{w}_{T+1}) \right] \right\} \tag{9}$$

where $b_t$ is the basin-wide net benefits at time $t$, $x_t$ the vector of allocation decisions, $w_t$ the vector of state variables, $\alpha$ a discount factor, $\nu$ a terminal value function, $\mathbb{E}$ the expectation operator capturing he uncertainty that governs the hydrologic inflow $q_t$ and $Z$ the total benefit associated with the optimal allocations $(x_1^*, x_2^*, ..., x_T^*)$.

This function is maximized to the extent permitted by physical, institutional or economic constraints :

$$g_{t+1}(\mathbf{x}_{t+1}) \leq 0 \tag{10}$$

$$h_{t+1}(\mathbf{w}_{t+1}) \leq 0 \tag{11}$$

$$\mathbf{w}_{t+1} = f_t(\mathbf{w}_t, \mathbf{x}_t, \mathbf{q}_t) \tag{12}$$

where $g$ is a set of functions constraining the allocation decision, $h$ a set of functions constraining the state of the system and $f$ a set of functions describing the transition of the system from time $t$ to time $t+1$.

Included in the functions in Eq. (12) are the mass balance equations for the river basin :

$$s_{t+1} - R(r_t + l_t) - I(i_t) + e_t(s_t, s_{t+1}) = s_t + q_t \tag{13}$$

where $s_t$ is the storage at time $t$, $r_t$ the controlled outflows, $l_t$ the uncontrolled outflows, $i_t$ the water withdrawals, $R$ and $I$ the connectivity matrices representing the topology of the system (including return flows), and $e_t$ the evaporation losses.

At the optimal solution of the problem (Eqs. (9) to (12)), the solver provides the allocation decisions $(x_1^*, x_2^*, ..., x_T^*)$ and the marginal values of water (shadow prices) $(\lambda_1, \lambda_2, ..., \lambda_T)$ of the constraints. For the constrains in Eq. (12), the shadow prices correspond to the marginal resource opportunity cost at the sites where water balances are computed.

*Acknowledgements.* The authors are grateful to Yasir Mohamed (HRC-Sudan) and Erik Ansink (Utrecht University) for valuable discussions early in the development of this work, and would like to thank the Institut Hydro-Québec en environnement, développement et société (EDS) for their financial support (Grant 03605-FO101829).

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
