# Peer review of "Sharing water and benefits in transboundary river basins"

_Hydrology and Earth System Sciences, 2016_

## Referee Comment (RC1) · Anonymous Referee #1 · 7 Mar 2016

General Comments:

The paper considers an institutional arrangement to distribute welfare in a river basin by maximizing the economic benefits of water use and then sharing these benefits using a (game theoretic?) method developed through stakeholder involvement. The methodology was applied to the Eastern Nile River basin.

The paper makes an interesting contribution to the body of knowledge surrounding calculating the benefits of transboundary water sharing. However, there are several shortcomings that should be addressed before the paper can be published in the journal. First the Methodology section of the paper is incomplete and needs to be improved as suggested in the specific comments below, mainly tha the axiomatic process that implements the bankruptcy game should be introduced and explained in the methodology section. Otherwise, the main potential contribution of the paper is without a methodological basis and is completely ad hoc depending on the site being studied. Second, the method was not actually applied using real stakeholders but it is applied to the widely studied Easter Nile Basin. The authors need to acknowledge the history of water use in this basin and how the benefits sharing indicated in the results of the paper differ from recent or projected use of water in the basin.

Specific Comments:

P.1-L.20: "There is a consensus among water professionals that the cooperative management of shared river basins should provide opportunities to increase the scope and scale of benefits" The authors have provided a single reference to justify this assertion. A broader consensus needs to be demonstrated before this statement can be accepted.

P.2-L.10: "water is allocated to maximize the net benefits from water use over the whole basin (economically efficient allocation)." Not all of these papers take the economist's position that one can simply maximize the benefits of water use in a basin and many of them recognize the political and administrative boundaries present in their case study basins and how those boundaries affect (restrict) the allocation of water in the basins.

P.3-L.30: "pseudo-market approach, a river basin authority (RBA) plays the role of water system operator, identifying economically efficient allocation policies which are then imposed on the agents (water users). The agents are charged for water, payments are redistributed to ensure equitability among the users." "the RBA collects information that is required to assess the demand curves, or at least the productivity of all users in the system, once at the beginning of each year." How realistic is this? In many parts of the world, this information is considered confidential. "...based on the bid information, the demand curve can be inferred using the residual imputation method..." This seems much more realistic that requiring users to give up their business information.

P.4-L. 15: "techniques such as remote sensing can be applied to validate land classification and cropping areas" Do the authors utilize these methods in this paper?

P.4-L.30: "allocation decisions are identified by matching demand with supply in a cost effective way, i.e. by giving priority of access to users with the highest productivity" It is not clear what the authors mean by "cost effective" way and this should be more clearly defined. Giving water to its highest valued use may be cost effective, but that depends on how you define "cost effective". Please clarify. As mentioned previously, this allocation method depends on the benevolent water manager having the authority to allocate the water is such a manner and in the real world this ignores any water rights or transboundary agreements that may exist in the basin. I think the authors should point out this limitation and discuss its implications in detail later in the paper.

P.4-L.30: "...power companies are considered non-rival water users since a unit of water released through one dam can be used downstream by another dam..." This may or may not be the case. In the case of the Syr Darya basin in Central Asia, this is certainly NOT the case since electricity production is in high demand in the winter when there is no irrigation water demand and hydropower releases in winter are lost to summer irrigation use. In the Eastern Nile, where the authors apply their model, the Grand Ethiopian Renaissance DamÂǎmay or may not be operated in a manner that allows the non-rival use of the water for power. The authors need to make this clear and explain the limitations of their assumptions.

P.6-L.5: "...Non-consumptive users buy inflow from the RBA, at the marginal value at the user site, and then sells the outflow downstream, back to the RBA, at the marginal value of water at the downstream site..." Why not just say that the users pay the difference between marginal value at the user site and the marginal value of water at the downstream site?

P.7-L.5: The Methodology section of the paper is incomplete since it does not indicate any method of determining the "transfer payments". The idea is stated that the "fairness" of the payments will be determined through an "axiomatic process" involving the

stakeholders, but no methodology is mentioned for how this procedure is carried out. Some description of a method should be given here, since this is the main contribution of the paper (the other components are well known and reported in the literature previously). Otherwise, the main potential contribution of the paper is without a methodological basis and is completely ad hoc depending on the site being studied. Section 3.5 presents much of the methodology (bankruptcy game theory) and should be moved back to Section 2 and the main aspects presented as general methodology.

P.13-L.5: "...for this study, the properties for this rule were not developed with stakeholder input as this was beyond the scope of this research project" So the method was not actually applied using real stakeholders. This fact needs to be pointed out in the abstract as it substantially weakens the impact of the paper. In addition, the authors do not acknowledge the history of water use in this river basin and the massive efforts that have been made to develop lasting and fair transboundary water sharing agreements in the basin. How do these historic efforts differ form the water allocation and benefits sharing indicated in the results of the authors' model? This should be explained and discussed in some detail, since this could be a major contribution of the paper to understanding water sharing in the Nile basin.

---

## Referee Comment (RC2) · Anonymous Referee #2 · 14 Mar 2016

**Arjoon et al. "Sharing water and benefits in transboundary river basins"**

**Summary**

This paper describes an approach to equitable sharing of benefits among multiple stakeholders in transboundary river basin systems. The basic idea is to maximize efficiency by allocating water where its value is highest, and then to collect payments from users using an axiomatic rule based on the marginal value of water at each site.

**General comments**

I like the paper and think it proposes an interesting approach. I see four main shortcomings that are not at all discussed, however. Ignoring them diminishes the credibility of the work.

1.  The first problem is that the issue of property rights is never discussed. Given how water rights are usually assigned in real world systems (and given their existence in the authors' application of interest, the Nile), this is a major problem.
2.  The second concern I have is about transaction costs. The need for a river basin authority to implement the allocation and sharing rule is taken for granted, and the cost of setting it up is totally ignored.
3.  The third concern is about perfect information (for the RBA). In a way, the authors fall directly into this trap with their rather simple assumptions about irrigation and hydropower values, by assuming that these are uniform across space and time in the Nile. It is almost certainly true that costs and productivity varies across sites, however. In general, this will greatly complicate the ability to establish an effective RBA for achieving the efficient and equitable allocation. Private information will also pose a problem, given that users at different locations in the basin have differing incentives to reveal their true valuations. The authors spend some time discussing preference revelation, but do not fully acknowledge the challenge.
4.  Finally, the approach depends on allowing all stakeholders a place at the table, but this seems unlikely. In the specific application, urban and environmental uses are imposed as constraints, which is one example of asymmetric bargaining position. There are likely other users that would be ignored as well.

**Specific comments**

Besides these three main comments, I have a few specific comments.

1.  The abstract makes it seem like there is no efficiency-equity tradeoff, but in general there is, except when a fully efficient compensation mechanism exists. The main paper acknowledges this more clearly.
2.  Introduction: Unidirectional flow is not what imposes externalities. Rephrase.
3.  Introduction: Is there really consensus that cooperative management increases benefits? Can you provide more than one citation to back this up? What about transaction costs?
4.  Can you discuss the implications of assuming constant marginal product of water in irrigation?

---

## Referee Comment (RC3) · Anonymous Referee #3 · 21 Apr 2016

This is a fine paper, but its contribution is somewhat hidden and not sufficiently developed/motivated. I have five comments that should help improve the paper.

1. P3 L 14–17. You implicitly state that axiomatic approaches ignore economic welfare. This is not exactly true. You may not be aware of some recent work in this area, e.g.: - Ambec, S., A. Dinar, and D. McKinney (2013). Water sharing agreements sustainable to reduced flows. Journal of Environmental Economics and Management 66(3), 639–655. - Van den Brink, R., G. van der Laan, and N. Moes (2012). Fair agreements for sharing international rivers with multiple springs and externalities. Journal of Environmental Economics and Management 63(3), 388–403. These papers apply axioms on the welfare distribution resulting from the physical allocation of water. Actually, most axiomatic papers in the river sharing literature do so.

[Figure]

2. You introduce, in Sections 2.1-2.3 a social planner that collects all information and derives an appropriate social cost of water and its related price. A tremendous task I would say, especially since water is not a regular good and this price will vary by quality, location, time, and possibly other aspects. What is more problematic is that the planner relies on all water users for its collection of information, a crucial step in the analysis. In Section 2.1 this process is described but this section ignores the problem posed by incentive compatibility: why would users truthfully reveal their demand curves (or make truthful bids) if they could benefit by pretending a higher demand curve (i.e. a higher bid)? Sure, the section mentions some methods to check the reliability of information, like remote sensing, but this does not eliminate the incentives to "cheat".

3. In general, it is not clear what the status quo / baseline situation is w.r.t. property rights over water, which makes it hard to interpret the model. I see three candidates for the status quo: - First, on P4 L26 an exogenous water price P_D is introduced. This suggests that there is a planner or a market active in the status quo, where users can buy their water. - Second, expected net benefits (ENB) are derived assuming that a user can abstract any water unhampered by other (upstream) users' water use. This suggests that you take the principle of Unlimited Territorial Integrity as your status quo. - Third, the sharing of the RBA money seems to ignore any historical water use rights. This suggests that the status quo is one without any water use or where the RBA owns all water (since apparently water prices are paid to the RBA).

4. In Section 2.1, ENB was calculated as consumer surplus. In Section 3.2, however, ENB is calculated as unconstrained water use (Dj) multiplied by productivity (Pj). This seems to be a completely different measure. Where consumer surplus equals willingness to pay minus the water price for all consumed units of water and is measured in money terms, this new measure is a production measure: productivity of water times consumed units of water, probably measured in terms of physical output. This is very confusing (it is also confusing that P is used to denote both productivity and price). In section 3.3, Eq (2), again the production measure is used to calculate gross benefits.

Gross benefits cannot be the product of water use and productivity.

5. The innovative part of the paper is where you distribute the rents using the axiomatic approach. This method is postponed to Section 3.5. My main comment here is that there is no clear motivation for distributing E such that each user obtains an equal proportion of benefits (FNB+tp) to claims (ENB). There are many axiomatic solutions that are similar in spirit to yours, but I do not see a compelling motivation why this particular new rule is introduced and applied here. It seems standard to motivate a new solution in terms of its characterizing properties, but such a characterization is not provided here. There are some statements in the text that claim this rule satisfies the properties "solidarity" and "security of minimum benefits", but these properties are not clearly defined. Note that I am not saying that a full characterization should be provided here, as that is perhaps less relevant for the HESS audience, but I would expect a convincing motivation for introducing this new solution over any other (existing) solutions. Two additional minor comments: - By taking account of FNB in your bankruptcy rule, you have a problem that is more general than a standard bankruptcy problem (see e.g. work by Hougaard). - Your proposed solution does not take into account historical water use or any other property rights regime? (see my comment on the status quo).

---

## Author Comment (AC1) · 4 May 2016

**"Sharing water and benefits in transboundary river basins" by D. Arjoon et al.**

The authors would like to thank the reviewer for his/her interesting and constructive comments. Our responses and the proposed changes/corrections are detailed below.

Referee #1

General Comments:

1.1 The paper considers an institutional arrangement to distribute welfare in a river basin by maximizing the economic benefits of water use and then sharing these benefits using a (game theoretic?) method developed through stakeholder involvement. The methodology was applied to the Eastern Nile River basin.

RESPONSE: In Section 2.4, we describe a method of sharing the economic benefits which "should be determined in collaboration with the water users. Properties that define fairness, as determined through negotiations with the water users, are then translated into a sharing rule using an axiomatic approach." This description of the sharing method is left intentionally very general, since this will be different for each river basin. How the benefits will be shared depends entirely on the definition of fairness that results from negotiations with the water users. Benefits could be shared proportionally or using an egalitarian method or some other form of sharing could be used. We have further clarified this in the first paragraph of Section 2.4 by adding the following statement: *"How the benefits are shared depends entirely on this definition as agreed to by water users. For example, a simple proportional sharing method may satisfy the properties of equity defined by the users, or an egalitarian method, or some other form of sharing may be required."*

In our case study, using the Eastern Nile River Basin (Section 3), the method used is not game theoretic; we abstract from any stability/equilibrium analysis. We do not investigate the possibility that one or more of the water users could be better off on their own. Instead, we propose a mechanism whereby overall benefits would be maximized as a result of full cooperation and then shared according to a key perceived as fair by the different water users. In other words, the proposed institutional arrangement makes sure that (1) the size of the pie is the largest and (2) the pie is shared in an equitable manner between the participants.

1.2 The paper makes an interesting contribution to the body of knowledge surrounding calculating the benefits of transboundary water sharing. However, there are several shortcomings that should be addressed before the paper can be published in the journal. First the Methodology section of the paper is incomplete and needs to be improved as suggested in the specific comments below, mainly that the axiomatic process that implements the bankruptcy game should be introduced and explained in the methodology section. Otherwise, the main potential contribution of the paper is without a methodological basis and is completely ad hoc depending on the site being studied. Second, the method was not actually applied using real stakeholders but it is applied to the widely studied Eastern Nile Basin. The authors need to acknowledge the history of water use in this basin and how the benefits sharing indicated in the results of the paper differ from recent or projected use of water in the basin.

RESPONSE:

1. We agree that the last part of the methodology section might look ad hoc. This is because the benefit sharing mechanism is meant to be flexible since it will depend on the specific conditions of the basin being studied. Section 2.4 has been changed to make this more clear. It now states: "*At this point in the methodology, the RBA has collected an amount of money, referred to as the estate (E), that can be shared among the water use agents. Using an axiomatic approach, a method of sharing this estate should be determined. The aim of the axiomatic approach is to find and capture the notion of fairness that water users could agree upon. The approach then sets out axioms (properties) that fairness should or should not satisfy. Finally, these properties are translated into a sharing rule that quantifies the particular definition of fairness. How the benefits are shared depends entirely on this definition as agreed to by water users. For example, a simple proportional sharing method may satisfy the properties of equity defined by the users, or an egalitarian method, or some other form of sharing may be required. Since each river basin will have a different definition of fairness (depending on conditions in the basin and the outcome of negotiations with the water users), each river basin will likely have its own unique sharing rule.*"

2. A brief history of water sharing agreements in the Nile River Basin is given in Section 3.1 of the paper. The purpose of the presented methodology is an alternative to these types of agreements on international river basins, which are often perceived as zero-sum games and can lead to distrust and tension between riparian countries, as is the case in the Nile River Basin. What we present is an entirely different perspective that may help to avoid the pitfalls and limitations of current agreements.  For example, with respect to the Nile Basin, the current agreement driving water allocation legally constrains Sudan to 18.5 bcm of water use. Sudan has land resources to expand irrigation and use much more water than this (Allan et al 2013), but is limited due to the agreement. As well, uncertainty with respect to changing climate and the possibility of increased evaporation, uncertain hydrology and sea level rise could create an imbalance in water demand and supply in the basin (Whittington, 2014). A rise in sea level would result in the loss of agricultural land in the Nile Delta and, subsequently, a large portion of Egypt's historic water use would no longer be required (Whittington, 2014). We have added a sentence to the last paragraph in the introduction (Section 1) which states "*The institutional arrangement described in this paper should encourage full cooperation between water users because it is intended as a replacement for traditional types of agreements on international river basins, which can lead to distrust and tension between riparian countries. What we present is an entirely different perspective that may help to avoid the pitfalls and limitations of current agreements.*"

   It is difficult to compare the results of the case study with current water use in the basin. The presented case study is highly hypothetical and is not consistent with the actual, current allocation scheme. In the case study, we assume complete cooperation, there is expanded irrigation in the basin and the Grand Ethiopian Renaissance Dam is online.  This represents a possible long-term future scenario in the basin and the results reflect this. We have added a paragraph to the end of the results (Section 4) to clarify this:  "*Finally, it should be noted that we make no attempt to compare the results of the case study with current water use in the basin. While the presented case study is hypothetical and is not consistent with the actual, current situation, it represents a possible long-term future scenario in the basin and the results reflect these assumptions. In the case study, we assume complete cooperation, there is expanded irrigation in the basin and the Grand Ethiopian Renaissance Dam is online.*"

1.3    P.1-L.20: "There is a consensus among water professionals that the cooperative management of shared river basins should provide opportunities to increase the scope and scale of benefits" The authors have provided a single reference to justify this assertion. A broader consensus needs to be demonstrated before this statement can be accepted.

RESPONSE: More references have been added. *"There is a consensus among water professionals that the cooperative management of shared river basins should provide opportunities to increase the scope and scale of benefits (Phillips et al., 2006; Grey and Sadoff, 2007; Leb, 2015), stepping beyond the volumetric allocation of water that reduces negotiations between riparians to a zero-sum game."*

1.4    P.2-L.10: "water is allocated to maximize the net benefits from water use over the whole basin (economically efficient allocation)." Not all of these papers take the economist's position that one can simply maximize the benefits of water use in a basin and many of them recognize the political and administrative boundaries present in their case study basins and how those boundaries affect (restrict) the allocation of water in the basins.

RESPONSE: We agree that not all of these papers take the position that the benefits of water use can simply be maximized without recognizing the various constraints within the cases studied. We have changed the wording in the 3rd paragraph of the introduction to make this clearer. The sentence now reads *"The traditional approach to estimating the economic benefits of cooperation relies on hydro-economic modelling (Arjoon et al., 2014; Jeuland et al., 2014; Tilmant and Kinzelbach, 2012; Teasley and McKinney, 2011; Whittington et al., 2005). These studies present various implementation strategies representing various levels of cooperation, but all show that there are significant economic benefits to be had through basin-wide cooperation."*

1.5    P.3-L.30: "pseudo-market approach, a river basin authority (RBA) plays the role of water system operator, identifying economically efficient allocation policies which are then imposed on the agents (water users). The agents are charged for water, payments are redistributed to ensure equitability among the users." "the RBA collects information that is required to assess the demand curves, or at least the productivity of all users in the system, once at the beginning of each year." How realistic is this? In many parts of the world, this information is considered confidential. ". . .based on the bid information, the demand curve can be inferred using the residual imputation method. . ." This seems much more realistic that requiring users to give up their business information.

RESPONSE: The authors believe that in the future, it will be realistic to get some of this information. Currently, market prices, either national or international, can be observed and transportation costs can be estimated, allowing for an approximation of the mark-up that may accrue to farmers, for example. We stress that this paper describes a system in which it is assumed that there is cooperation over the whole basin. This means that water users have agreed to bid for water and to supply the information that is necessary to make the methodology work. It is up to the RBA to check that the information is reliable. Increasingly, river basins are being monitored and the information required is becoming available (for example, current hydromet projects in the Senegal River Basin). The system may not seem realistic at this point,

but, in the long-term, exchange of information will increase the availability of data over river basins. This increase in information exchange is in keeping with the obligation to cooperate and exchange information that is outlined in the UN Convention on the Law of the Non-navigational Uses of International Watercourses.

We have added a paragraph to the conclusions Section 5 (second to last paragraph) to discuss the constraint of available/reliable data. This paragraph reads: "*Another constraint is the availability of reliable data. Some information such as market prices, either national or international, can be observed and transportation costs can be estimated, allowing for an approximation of the mark-up that may accrue to farmers, for example. This paper describes a system in which it is assumed that there is cooperation over the whole basin and that water users have agreed to bid for water and to supply the information that is necessary to make the methodology work. Increasingly, the information required is becoming available through the use of remote sensing and monitoring of river basins.*"

1.6    P.4-L. 15: "techniques such as remote sensing can be applied to validate land classification and cropping areas" Do the authors utilize these methods in this paper?

RESPONSE: It is up to the RBA to check that the information given by water users is reliable. Remote sensing is one of the techniques available to validate information such as land classification and cropping areas. We have updated paragraph 2 in Section 2.1 to more clearly state this. This paragraph now states: "*In order to control the declarations of agents in the agricultural sector, the RBA can use techniques such as remote sensing to validate land classification and cropping areas (Gallego et al., 2014; El-Kawy et al., 2011; Rozenstein and Karnieli, 2011).*"

We do not use these methods in the present case study.

1.7    P.4-L.30: "allocation decisions are identified by matching demand with supply in a cost effective way, i.e. by giving priority of access to users with the highest productivity" It is not clear what the authors mean by "cost effective" way and this should be more clearly defined. Giving water to its highest valued use may be cost effective, but that depends on how you define "cost effective". Please clarify. As mentioned previously, this allocation method depends on the benevolent water manager having the authority to allocate the water is such a manner and in the real world this ignores any water rights or transboundary agreements that may exist in the basin. I think the authors should point out this limitation and discuss its implications in detail later in the paper.

RESPONSE:

1.  The authors have changed the term "cost-effective" to "cost-efficient" implying least cost, or maximum productivity.

2.  The allocation method departs from traditional (physical) allocation mechanisms based on water rights and relies instead on a bidding process whereby all water users are granted equal access to the resource. Productive use and allocation decisions are separated. The benevolent water manager is a non-profit, regulated organization that acts as a third party operator of the water resources system. In other words, it does not directly put water to productive use for its own benefit. Instead, it coordinates allocation decisions throughout the system based on the offers provided by eligible water users, and tries to achieve allocative

efficiency by ensuring that the good or service is consumed by those who value it most highly. The benevolent water manager, then, is the operator of an auction-based market. We agree that this is a highly hypothetical scenario but technological changes (e.g. availability of massive remote sensing data) combined with the need to achieve greater efficiency due to external pressures (population growth, climate change) might trigger major regulatory reforms in the water sector. This was seen in the energy sector in the late XXth century where, before 1970, energy generation was widely believed to be part of a natural monopoly. Technological developments such as cheap gas-fired power plants, combined with costly and inefficient investments made by the monopolies, suggested that competition was needed and lead to the introduction of deregulated electricity markets. This manuscript must be seen as a prospective analysis. We are concerned with a future situation that does not currently exist and we look at how the institutional arrangement would perform under these conditions.

1.8     P.4-L.30: "...power companies are considered non-rival water users since a unit of water released through one dam can be used downstream by another dam. . ." This may or may not be the case. In the case of the Syr Darya basin in Central Asia, this is certainly NOT the case since electricity production is in high demand in the winter when there is no irrigation water demand and hydropower releases in winter are lost to summer irrigation use. In the Eastern Nile, where the authors apply their model, the Grand Ethiopian Renaissance DamÂaˇmay or may not be operated in a manner that allows the non-rival use of the water for power. The authors need to make this clear and explain the limitations of their assumptions.

RESPONSE: Thank you for this insight. We are in agreement. We have changed the text in this section to reflect this. The first paragraph in Section 2.2 now includes the following statement: "*For example, water flowing through a dam may be considered a non-rival water use since a unit of water released through one dam can be used downstream by another dam. In rival water use, units are consumed and are no longer available to other water users (for example, water lost to irrigation or water held in a reservoir during a period when it is required downstream for irrigation).*"

1.9     P.6-L.5: ". . .Non-consumptive users buy inflow from the RBA, at the marginal value at the user site, and then sells the outflow downstream, back to the RBA, at the marginal value of water at the downstream site. . ." Why not just say that the users pay the difference between marginal value at the user site and the marginal value of water at the downstream site?

RESPONSE: This sentence now reads "*Non-consumptive users buy inflow from the RBA at a price equal to the difference between the marginal value of water at the user site and the marginal value of water at the downstream site (Fig. 3).*"

1.10    P.7-L.5: The Methodology section of the paper is incomplete since it does not indicate any method of determining the "transfer payments". The idea is stated that the "fairness" of the payments will be determined through an "axiomatic process" involving the stakeholders, but no methodology is mentioned for how this procedure is carried out. Some description of a method should be given here, since this is the main contribution of the paper (the other components are well known and reported in the literature previously). Otherwise, the main potential contribution of the paper is without a methodological basis and is completely ad hoc depending

on the site being studied. Section 3.5 presents much of the methodology (bankruptcy game theory) and should be moved back to Section 2 and the main aspects presented as general methodology.

RESPONSE: We have updated the first paragraph in Section 2.4 to further describe the method of transfer payments. We have added the following: "*How the benefits are shared depends entirely on this definition as agreed to by water users. For example, a simple proportional sharing method may satisfy the properties of equity defined by the users, or an egalitarian method, or some other form of sharing may be required.*"

Please see the response to comments 1.1 and 1.2 for further details.

1.11    P.13-L.5: ". . .for this study, the properties for this rule were not developed with stakeholder input as this was beyond the scope of this research project" So the method was not actually applied using real stakeholders. This fact needs to be pointed out in the abstract as it substantially weakens the impact of the paper. In addition, the authors do not acknowledge the history of water use in this river basin and the massive efforts that have been made to develop lasting and fair transboundary water sharing agreements in the basin. How do these historic efforts differ form the water allocation and benefits sharing indicated in the results of the authors' model? This should be explained and discussed in some detail, since this could be a major contribution of the paper to understanding water sharing in the Nile basin.

RESPONSE:

1.  While it is true that the method for determining transfer payments was not developed using stakeholder input, we do not believe that this weakens the impact of the methodology. We have an objective point of view and our analysis is a benchmark or reference point. We have updated the paper to include a mention of this in the text. A paragraph has been added to Section 3.5 which reads "*It should be noted that, for this study, the properties for this rule were not developed with stakeholder input as this was beyond the scope of this research project. Although stakeholder involvement is imperative in this institutional arrangement, in this case study, we are giving an objective viewpoint and this analysis serves as a benchmark or reference point.*"

2.  Please see the response to comment 1.2.2 for a detailed response to the question of water sharing agreements in the basin.

Allan, J. A., Keulertz, M., Sojamo, S. & Warner, J. eds. (2013). Handbook of Land and Water Grabs in Africa: Foreign Direct Investment and Food and Water Security. Routledge International Handbook. Routledge, Abingdon.

Whittington, D, J. Waterbury, and M. Jeuland (2014), The Grand Renaissance Dam and prospects for cooperation on the Eastern Nile, *Water Policy*, 16, 595–608.

---

## Author Comment (AC2) · 4 May 2016

**"Sharing water and benefits in transboundary river basins" by D. Arjoon et al.**

The authors would like to thank the reviewer for his/her interesting and constructive comments. Our responses and the proposed changes/corrections are detailed below.

Referee #2

Summary

This paper describes an approach to equitable sharing of benefits among multiple stakeholders in transboundary river basin systems. The basic idea is to maximize efficiency by allocating water where its value is highest, and then to collect payments from users using an axiomatic rule based on the marginal value of water at each site.

General comments

I like the paper and think it proposes an interesting approach. I see four main shortcomings that are not at all discussed, however. Ignoring them diminishes the credibility of the work.

2.1     The first problem is that the issue of property rights is never discussed. Given how water rights are usually assigned in real world systems (and given their existence in the authors' application of interest, the Nile), this is a major problem.

RESPONSE: A brief history of water sharing agreements in the Nile River Basin is given in Section 3.1 of the paper. The purpose of the presented methodology is as an alternative to these types of agreements on international river basins, which can lead to distrust and tension between riparian countries, as is the case in the Nile River Basin. What we present is an entirely different perspective that may help to avoid the pitfalls and limitations of current agreements. For example, with respect to the Nile Basin, the current agreement driving water allocation legally constrains Sudan to 18.5 bcm of water use. Sudan has land resources to expand irrigation and use much more water than this (Allan et al 2013), but is limited due to the agreement. As well, uncertainty with respect to changing climate and the possibility of increased evaporation, uncertain hydrology and sea level rise could create an imbalance in water demand and supply in the basin (Whittington, 2014). A rise in sea level would result in the loss of agricultural land in the Nile Delta, and, subsequently, a large portion of Egypt's historic water use would no longer be required (Whittington, 2014). We have added a sentence to the last paragraph in the introduction (Section 1) which states "*The institutional arrangement described in this paper should encourage full cooperation between water users because it is intended as a replacement for traditional types of agreements on international river basins, which can lead to distrust and tension between riparian countries. What we present is an entirely different perspective that may help to avoid the pitfalls and limitations of current agreements.*"

2.2     The second concern I have is about transaction costs. The need for a river basin authority to implement the allocation and sharing rule is taken for granted, and the cost of setting it up is totally ignored.

RESPONSE: As stated in P21-L9 "One obvious constraint of this method is its dependance on the existence of a strong basin-wide authority...". The assumption is made that the RBA exists. This is not unrealistic given that there are a number of river basins that already have an RBA in place (OMVS on the Senegal, MRC on the Mekong, ZAMCOM on the Zambezi, NBA on the

Niger River, etc.) and others that are working toward this goal (Volta Basin Authority, for example). As well, in this methodology, we assume that the countries cooperate through the RBA and we agree that this involves transaction costs. If countries agree to the kind of institutional arrangement described, they do so because these transaction costs are less than the cost of cooperation. As well, the transaction cost is not proportional to water allocation or use and, hence, could be introduced as a fixed cost, as we have done in the case study. A fixed cost would diminish the estate available to share and, ultimately, the final benefits of the water users, but it would not alter the proportion.

2.3     The third concern is about perfect information (for the RBA). In a way, the authors fall directly into this trap with their rather simple assumptions about irrigation and hydropower values, by assuming that these are uniform across space and time in the Nile. It is almost certainly true that costs and productivity varies across sites, however. In general, this will greatly complicate the ability to establish an effective RBA for achieving the efficient and equitable allocation. Private information will also pose a problem, given that users at different locations in the basin have differing incentives to reveal their true valuations. The authors spend some time discussing preference revelation, but do not fully acknowledge the challenge.

RESPONSE: We agree that simple assumptions about irrigation and hydropower values in the case study were made.  These assumptions have been made by other researchers (Whittington, 2005) and are generally consistent with international experience in well-run irrigation schemes and power systems. However, it is obvious that the quality of the results are dependent on the quality and availability of data to run the models, and on the assumptions made. This is a challenge in all studies of this type.

The authors believe that, in the future, it will be realistic to get some of this data. Currently, market prices, either national or international ones, can be observed and transportation costs can be estimated, allowing for an approximation of the mark-up that may accrue to farmers, for example. We stress that this paper describes a system in which it is assumed that there is cooperation over the whole basin.  This means that water users have agreed to bid for water and to supply the information that is necessary to make the methodology work. It is up to the RBA to check that the information is reliable. Increasingly, river basins are being monitored and the information required is becoming available (for example, current  hydromet projects in the Senegal River Basin). The system may not seem realistic at this point but, in the long-term, exchange of information will increase the availability of data over river basins. This increase in information exchange is in keeping with the obligation to cooperate and exchange information that is outlined in the UNConvention on the Law of the Non-navigational Uses of International Watercourses.

We have added a paragraph to the conclusions Section 5 (second to last paragraph) to discuss the constraint of available/reliable data. This paragraph reads: "*Another constraint is the availability of reliable data. Some information such as market prices, either national or international ones, can be observed and transportation costs can be estimated, allowing for an approximation of the mark-up that may accrue to farmers, for example. This paper describes a system in which it is assumed that there is cooperation over the whole basin and that water users have agreed to bid for water and to supply the information that is necessary to make the methodology work. Increasingly, the information required is becoming available through the use of remote sensing and monitoring of river basins.*"

2.4 Finally, the approach depends on allowing all stakeholders a place at the table, but this seems unlikely. In the specific application, urban and environmental uses are imposed as constraints, which is one example of asymmetric bargaining position. There are likely other users that would be ignored as well.

RESPONSE: Allowing all stakeholders a place at the table is indeed challenging, especially for large systems with diversified water use activities. In the irrigation sector, for instance, farmers could send a representative, e.g. a member of the water user association. For uses of water as a public good (e.g. environmental flows), the representative could be the Ministry of Environment of the country of interest. For municipal uses, the system could be designed in such a way that a minimum amount of allocated water is guaranteed (a fixed constraint in the allocation system) while quantities beyond that minimum would be part of the pool for which municipalities would have to bid. Industrial and power companies are easier to handle. All users that can be rationed (mainly private water users) are allowed a place at the table for the purpose of defining fairness with respect to transfer payments.

Another possibility is that the government (or at least a high level representative of the stakeholders) has the ultimate negotiation power, akin to negotiations on trade liberalizations. Clearly, different lobbies exist that would try to influence the government, implying, ultimately, some form of compensation (the analysis of which would lie outside the scope of this paper).

Specific comments

Besides these three main comments, I have a few specific comments.

2.5 The abstract makes it seem like there is no efficiency-equity tradeoff, but in general there is, except when a fully efficient compensation mechanism exists. The main paper acknowledges this more clearly.

RESPONSE: With the physical allocation of water, policy makers face an efficiency-equity trade-off. With the proposed benefit-sharing mechanism, the trade-off still exists but the extent of the imbalance between the two is reduced because benefits are maximized and redistributed according to a key that has been collectively agreed on by the participants. We have added the following sentence to the abstract to highlight this: *"With the proposed benefit-sharing mechanism, the efficiency-equity trade-off still exists but the extent of the imbalance is reduced because benefits are maximized and redistributed according to a key that has been collectively agreed upon by the participants."*

2.6 Introduction: Unidirectional flow is not what imposes externalities. Rephrase.

RESPONSE: This sentence in the introduction (Section 1) has been rephrased to read: *"This is particularly evident in the case of transboundary river basins in which unidirectional, negative externalities, caused by the upstream regulation of the natural flow, often place some parties at a disadvantage and results in asymmetric relationships that add to the challenge of coordinating resource use (van der Zaag, 2007)."*

2.7 Introduction: Is there really consensus that cooperative management increases benefits? Can you provide more than one citation to back this up? What about transaction costs?

RESPONSE: Additional citations have been added. The statement now reads: "*There is a consensus among water professionals that the cooperative management of shared river basins should provide opportunities to increase the scope and scale of benefits (Phillips et al., 2006; Grey and Sadoff, 2007; Leb, 2015), stepping beyond the volumetric allocation of water that reduces negotiations   between riparians to a zero-sum game.*"

2.8      Can you discuss the implications of assuming constant marginal product of water in irrigation?

RESPONSE: In our case study there are 3 (upstream) irrigation schemes that do not (on average) get the amount of water demanded. This is because the productivity of water that is used over the entire basin (0.05 USD/m3) is less than the marginal value of water at these nodes in the system. This productivity value is used because there is currently a lack of data for irrigation in the basin. The availability of economic/agricultural data in each irrigation scheme over the basin, as well as details of cropping patterns in each scheme, would allow us to develop a non-horizontal demand curve. If this were the case, high value crops  in the upstream schemes may be irrigated and the low value crops in downstream schemes may not be irrigated.  This means that the irrigation water users that are rationed may change and may be spread out over the basin. The first paragraph in the results section (Section 4) has been updated to the following statement: "*The rationing of water for upstream irrigation users is a result of the horizontal demand curve used for irrigation. If more detailed economic/agricultural data were available, a non-horizontal demand curve could be produced. This may result in irrigation schemes with high value crops having priority to water and those areas with low value crops not being irrigated. This means that the irrigation water users that are rationed may change and they may be more spread out over the basin.*"

---

## Author Comment (AC3) · 4 May 2016

**"Sharing water and benefits in transboundary river basins" by D. Arjoon et al.**

The authors would like to thank the reviewer for his/her interesting and insightful comments. Our responses and the proposed changes/corrections are detailed below.

Referee #3

3.1    P3 L 14–17. You implicitly state that axiomatic approaches ignore economic welfare. This is not exactly true. You may not be aware of some recent work in this area, e.g.: - Ambec, S., A. Dinar, and D. McKinney (2013). Water sharing agreements sustainable to reduced flows. Journal of Environmental Economics and Management 66(3), 639– 655. - Van den Brink, R., G. van der Laan, and N. Moes (2012). Fair agreements for sharing international rivers with multiple springs and externalities. Journal of Environ- mental Economics and Management 63(3), 388–403. These papers apply axioms on the welfare distribution resulting from the physical allocation of water. Actually, most axiomatic papers in the river sharing literature do so.

RESPONSE: We do not agree that we implicitly abstract from economic welfare. The text in question reads: "As discussed previously, the economically efficient allocation of water is not necessarily equitable. Conversely, axiomatic approaches may be considered equitable but do not necessarily maximize the total economic welfare over the basin and may be considered deficient as a result. Institutional arrangements that ensure maximum economic welfare, as well as the equitable sharing of these benefits over the basin, are required."

We use the word "deficient" to mean that the axiomatic approaches may result in less than optimal water allocations from an economic perspective. For example, Madani et. al. (2014) uses bankruptcy rules to determine the allocation of water within the Qezelozan-Sefidrud river system in Iran. The resulting allocations are defined by the notion of fairness that are inherent in each rule, but these rules do not necessarily maximize the economic welfare over the basin.

In order to clarify this, we have changed the paragraph to read "*As discussed previously, the economically efficient allocation of water is not necessarily equitable. Axiomatic approaches, on the other hand, allow the characterization of an equitable distribution of welfare, but do not necessarily maximize the aggregated economic welfare over the basin. Institutional arrangements that ensure maximum economic welfare, as well as the equitable sharing of these benefits over the basin, are required.*"

3.2    You introduce, in Sections 2.1-2.3 a social planner that collects all information and derives an appropriate social cost of water and its related price. A tremendous task I would say, especially since water is not a regular good and this price will vary by quality, location, time, and possibly other aspects. What is more problematic is that the planner relies on all water users for its collection of information, a crucial step in the analysis. In Section 2.1 this process is described but this section ignores the problem posed by incentive compatibility: why would users truthfully reveal their demand curves (or make truthful bids) if they could benefit by pretending a higher demand curve (i.e. a higher bid)? Sure, the section mentions some methods to check the reliability of information, like remote sensing, but this does not eliminate the incentives to "cheat".

RESPONSE: We agree that the incentives to cheat will remain even if the river basin authority is able to audit the bids. For industrial uses, including hydropower generation, cheating might be more difficult because the market prices and production functions are often well characterized. In our opinion, the main challenge is to be found in the agricultural sector because (a) it is often the largest water use (and hence cheating might have serious basin-wide consequences), and (b) the heterogeneity in terms of cropping patterns and irrigation efficiency requires that significant data be collected and analyzed to audit the demands. However, due to river basin closure, there is a strong incentive to strengthen the monitoring of river basins, either directly (on-site measurement stations) or indirectly (remote sensing). Various initiatives, at different levels, demonstrate that significant effort and financial resources are being devoted to observations of water resources. For example, the Surface Water and Ocean Topography (SWOT) satellite mission (anticipated launch date 2020), The Sentinel-3 satellite mission, the Hydromet project in the Senegal River basin, etc. We argue that the incentives to cheat might not be eliminated but they can be suppressed, or at least kept within limits, through a robust monitoring system and a strong RBA to negotiate disputes.  An example of how this has worked, with good success, is the Indus River basin. Zawahri (2009), in discussing the Permanent Indus Commission, states "The commission's ability to monitor development of the shared river system has permitted it to ease member states' fear of cheating and confirm the accuracy of all exchanged data. Finally, its conflict resolution mechanisms have permitted the commission to negotiate settlements to disputes and prevent defection from cooperation."

3.3    In general, it is not clear what the status quo / baseline situation is w.r.t. property rights over water, which makes it hard to interpret the model. I see three candidates for the status quo: - First, on P4 L26 an exogenous water price $P\_D$ is introduced. This suggests that there is a planner or a market active in the status quo, where users can buy their water.  Second, expected net benefits (ENB) are derived assuming that a user can abstract any water unhampered by other (upstream) users' water use. This suggests that you take the principle of Unlimited Territorial Integrity as your status quo. - Third, the sharing of the RBA money seems to ignore any historical water use rights. This suggests that the status quo is one without any water use or where the RBA owns all water (since apparently water prices are paid to the RBA).

RESPONSE:  The purpose of the presented methodology is to provide an alternative to the types of agreements on international river basins which attempt to define the rights to water. These agreements are often perceived as zero-sum games and can lead to distrust and tension between riparian countries, as is the case in the Nile River Basin. What we present is an entirely different perspective that may help to avoid the pitfalls and limitations of current agreements based on physical allocation. For example, with respect to the Nile Basin, the current agreement driving water allocation legally constrains Sudan to 18.5 bcm of water use. Sudan has available land resources to expand irrigation and use much more water than this (Allan et al 2013), but is limited due to the agreement. As well, uncertainty with respect to changing climate and the possibility of increased evaporation, uncertain hydrology and sea level rise could create an imbalance in water demand and supply in the basin (Whittington, 2014). For instance, a rise in sea level would result in the loss of agricultural land in the Nile Delta, and, subsequently, a large portion of Egypt's historic water use would no longer be required (Whittington, 2014). Therefore, as part of the application of this methodology to a river basin, the historical

water use rights are disregarded.

The institutional arrangement that we describe departs from traditional (physical) allocation mechanisms that are based on water rights and relies instead on a bidding process whereby all water users are granted equal access to the resource. Productive use and allocation decisions are separated. The benevolent water manager (RBA) is a non-profit, regulated organization that acts as a third party operator of the water resources system. It does not directly put water to productive use for its own benefit. Instead, it coordinates allocation decisions throughout the system based on the offers provided by water users, and tries to achieve allocative efficiency by ensuring that the water is consumed by those who value it most highly. The benevolent water manager is, in this case, the operator of an auction-based market. So, as part of the institutional arrangement, the RBA may be considered as the owner of bulk (raw) water in the basin. Since the RBA is a supranational institution, the riparian countries own the water. However, once the allocated water is diverted to the user, the water belongs to the user (who has paid for it). Note that price (P_D) is not exogenous; it is derived from the aggregate demand curve for water that results from the market operated by the RBA as part of the methodology.

As mentioned earlier, water users are invited to communicate basic economic information required to estimate their demand curve and to derive the expected net benefit (ENB), i.e. the benefit they would get without rationing. At this stage, we do assume that a user's benefits are maximized unhampered by other upstream users' water use or by the historical claims of downstream users. In a sense, the status quo, in this case, is a balance between two extreme principles: the principle of Unlimited Territorial Integrity and the principle of Absolute Territorial Sovereignty.

3.4    In Section 2.1, ENB was calculated as consumer surplus. In Section 3.2, however, ENB is calculated as unconstrained water use (Dj) multiplied by productivity (Pj). This seems to be a completely different measure. Where consumer surplus equals willing- ness to pay minus the water price for all consumed units of water and is measured in money terms, this new measure is a production measure: productivity of water times consumed units of water, probably measured in terms of physical output. This is very confusing (it is also confusing that P is used to denote both productivity and price). In section 3.3, Eq (2), again the production measure is used to calculate gross benefits. Gross benefits cannot be the product of water use and productivity.

RESPONSE: In section 2.1 of the paper, in Figure 1, we show the ENB as being the consumer surplus. Figure 1 is the demand function for conditional factors needed to produce a certain level of output. In our case, this is the demand for water needed to produce a certain amount of crop and there is unconstrained output. We disregard the fact that the WTP is, in fact, constrained by the final output level that one wishes to produce. In our case study (section 3), we implicitly assume that the input demand is horizontal (perfectly elastic) with the price (P) = marginal productivity. Underlying this assumption we suppose that the productivity remains the same for the producer and that the output can always be sold on the market. The gross consumer valuation is equal to the rectangle under the horizontal demand curve (or marginal productivity (USD/m3) x water quantity (m3)). For the ENB this is the whole area under the horizontal curve where the water quantity is equal to the water demand. For the gross benefits (GB), this is the area under the horizontal curve given the amount of water they are allocated. The final net benefits (FNB) are the GB minus the area under the horizontal curve representing the cost of

water. Please see the figure below which we will incorporate into the paper in section 3.

[Figure]

3.5     The innovative part of the paper is where you distribute the rents using the axiomatic approach. This method is postponed to Section 3.5. My main comment here is that there is no clear motivation for distributing E such that each user obtains an equal proportion of benefits (FNB+tp) to claims (ENB). There are many axiomatic solutions that are similar in spirit to yours, but I do not see a compelling motivation why this particular new rule is introduced and applied here. It seems standard to motivate a new solution in terms of its characterizing properties, but such a characterization is not provided here. There are some statements in the text that claim this rule satisfies the properties "solidarity" and "security of minimum benefits", but these properties are not clearly defined. Note that I am not saying that a full characterization should be provided here, as that is perhaps less relevant for the HESS audience, but I would expect a convincing motivation for introducing this new solution over any other (existing) solutions. Two additional minor comments: - By taking account of FNB in your bankruptcy rule, you have a problem that is more general than a standard bankruptcy problem (see e.g. work by Hougaard). - Your proposed solution does not take into account historical water use or any other property rights regime? (see my comment on the status quo).

RESPONSE: Please note our response to comment 3.3 regarding water rights.

As previously mentioned, section 2 describes the methodology while section 3 is an example of how this methodology can be applied using the Eastern Nile River Basin as the case study. In section 2.4 we describe the last step – transfer payments. We have added to this section, which should read: "*At this point in the methodology, the RBA has collected an amount of money, referred to as the estate (E), that can be shared among the water use agents. Using an axiomatic approach, a method of sharing this estate should be determined. The aim of the axiomatic approach is to find and capture the notion of fairness that water users could agree upon. The approach then sets out axioms (properties) that fairness should or should not satisfy. Finally, these properties are*

*translated into a sharing rule that quantifies the particular definition of fairness. How the benefits are shared depends entirely on the definition of fairness as agreed to by water users. For example, a simple proportional sharing method may satisfy the properties of equity defined by the users, or an egalitarian method, or some other form of sharing may be required. Since each river basin will have a different definition of fairness (depending on conditions in the basin and the outcome of negotiations with the water users), each river basin will likely have its own unique sharing rule."*

In section 3.5 we describe a possible solution to transfer payments, assuming that the agents have all agreed on the properties underlying this rule. It is important to note that there were no negotiations done to develop this rule (this was beyond the scope of the project), however, we do not believe that this weakens the impact of the methodology. We present an objective viewpoint and consider our analysis to be a benchmark or reference point. A paragraph has been added to section 3.5 which reads "*It should be noted that, for this study, the properties for this rule were not developed with stakeholder input as this was beyond the scope of this research project. Although stakeholder involvement is imperative in this institutional arrangement, in this case study, we are giving an objective viewpoint and this analysis serves as a benchmark or reference point.*"

The motivation for using this rule is that the cost of cooperation is divided equally among the agents. Again, we are certainly not saying that this solution is better than another, or even that this would be the solution to sharing the benefits for the Eastern Nile River Basin. Rather, we are giving an example of how the overall methodology could work when applied to a river basin. The rule or method used for the transfer payments would be based on the definition and properties of fairness that are developed through negotiations with the water users.

Allan, J. A., Keulertz, M., Sojamo, S. & Warner, J. eds. (2013). Handbook of Land and Water Grabs in Africa: Foreign Direct Investment and Food and Water Security. Routledge International Handbook. Routledge, Abingdon.

K. Madani, M. Zarezadeh, and S. Morid. A new framework for resolving conflicts over transboundary rivers us- ing bankruptcy methods. Hydrol. Earth Syst. Sci., 18: 3055–3068, 2014.

D.J. Whittington, D, J. Waterbury, and M. Jeuland. The Grand Renaissance Dam and prospects for cooperation on the Eastern Nile, *Water Policy*, 16, 595–608, 2014

N. Zawahri. India, Pakistan and cooperation along the Indus River system. Water Policy 11: 1-20, 2009